# Optimal compensation for neuron loss

**David GT Barrett[1,2†], Sophie Denève[1], Christian K Machens[2]\***

[1]Laboratoire de Neurosciences Cognitives, École Normale Supérieure, Paris, France;
[2]Champalimaud Neuroscience Programme, Champalimaud Centre for the Unknown, Lisbon, Portugal

**Abstract** The brain has an impressive ability to withstand neural damage. Diseases that kill neurons can go unnoticed for years, and incomplete brain lesions or silencing of neurons often fail to produce any behavioral effect. How does the brain compensate for such damage, and what are the limits of this compensation? We propose that neural circuits instantly compensate for neuron loss, thereby preserving their function as much as possible. We show that this compensation can explain changes in tuning curves induced by neuron silencing across a variety of systems, including the primary visual cortex. We find that compensatory mechanisms can be implemented through the dynamics of networks with a tight balance of excitation and inhibition, without requiring synaptic plasticity. The limits of this compensatory mechanism are reached when excitation and inhibition become unbalanced, thereby demarcating a recovery boundary, where signal representation fails and where diseases may become symptomatic.

## Introduction

The impact of neuron loss on information processing is poorly understood (*Palop et al., 2006*; *Montague et al., 2012*). Chronic diseases such as Alzheimer's disease cause a considerable decline in neuron numbers (*Morrison and Hof, 1997*), yet can go unnoticed for years. Acute events such as a stroke and traumatic brain injury can kill large numbers of cells rapidly, yet the vast majority of strokes are 'silent' (*Leary and Saver, 2003*). Similarly, the partial lesion or silencing of a targeted brain area may 'fail' to produce any measurable, behavioral effect (*Li et al., 2016*). This resilience of neural systems to damage is especially impressive when compared to man-made computer systems that typically lose all function following only minor destruction of their circuits. A thorough understanding of the interplay between neural damage and information processing is therefore crucial for our understanding of the nervous system, and may also help in the interpretation of various experimental manipulations such as pharmacological silencing (*Aksay et al., 2007*; *Crook and Eysel, 1992*), lesion studies (*Keck et al., 2008*), and optogenetic perturbations (*Fenno et al., 2011*).

In contrast to the study of information representation in damaged brains, there has been substantial progress in our understanding of information representation in healthy brains (*Abbott, 2008*). In particular, the theory of efficient coding, which states that neural circuits represent sensory signals optimally given various constraints (*Barlow, 1961*; *Atick, 1992*; *Olshausen and Field, 1996*; *Rieke et al., 1997*; *Simoncelli and Olshausen, 2001*; *Salinas, 2006*), has successfully accounted for a broad range of observations in a variety of sensory systems, in both vertebrates (*Simoncelli and Olshausen, 2001*; *Atick, 1992*; *Olshausen and Field, 1996*; *Smith and Lewicki, 2006*; *Greene et al., 2009*) and invertebrates (*Rieke et al., 1997*; *Fairhall et al., 2001*; *Machens et al., 2005*). However, an efficient representation of information is of little use if it cannot withstand some perturbations, such as normal cell loss. Plausible mechanistic models of neural computation should be able to withstand the type of perturbations that the brain can withstand.

In this work, we propose that neural systems maintain stable signal representations by optimally compensating for the loss of neurons. We show that this compensation does not require synaptic

**\*For correspondence:** christian.machens@neuro.fchampalimaud.org

**Present address:** [†]DeepMind, London, United Kingdom

**Competing interests:** The authors declare that no competing interests exist.

plasticity, but can be implemented instantaneously by a tightly balanced network whose dynamics and connectivity are tuned to implement efficient coding (*Boerlin et al., 2013*; *Denève and Machens, 2016*). When too many cells disappear, the balance between excitation and inhibition is disrupted, and the signal representation is lost. We predict how much cell loss can be tolerated by a neural system and how tuning curves change shape following optimal compensation. We illustrate these predictions using three specific neural systems for which experimental data before and after silencing are available – the oculomotor integrator in the hindbrain (*Aksay et al., 2000*, *2007*), the cricket cercal system (*Theunissen and Miller, 1991*; *Mizrahi and Libersat, 1997*), and the primary visual cortex (*Hubel and Wiesel, 1962*; *Crook and Eysel, 1992*; *Crook et al., 1996*, *1997*, *1998*). In addition, we show that many input/output non-linearities in the tuning of neurons can be re-interpreted to be the result of compensation mechanisms within neural circuits. Therefore, beyond dealing with neuronal loss, the proposed optimal compensation principle expands the theory of efficient coding and provides important insights and constraints for neural population codes.

## Results

### The impact of neuron loss on neural representations

We begin by studying how neuron loss can influence neural representations. We assume that information is represented within neural systems such that it can be read out or decoded by a downstream area through a linear combination of neural firing rates. More specifically, we imagine a system of $N$ neurons that represents a set of signals $\mathbf{x}(t) = (x_1(t), x_2(t), \ldots, x_M(t))$. These signals may be time-dependent sensory signals such as visual images or sounds, for instance, or, more generally, they may be the result of some computation from within a neural circuit. We assume that these signals can be read out from a weighted summation of the neurons' instantaneous firing rates, $r_i(t)$, such that

$$\hat{\mathbf{x}}(t) = \sum_{i=1}^{N} \mathbf{D}_i r_i(t),$$  (1)

where $\hat{\mathbf{x}}(t)$ is the signal estimate and $\mathbf{D}_i$ is a $M$-dimensional vector of decoding weights that determines how the firing of the $i$-th neuron contributes to the readout. Such linear readouts are used in many contexts and broadly capture the integrative nature of dendritic summation.

What happens to such a readout if one of the neurons is suddenly taken out of the circuit? A specific example is illustrated in *Figure 1A*. Here the neural system consists of only two neurons that represent a single, scalar signal, $x(t)$, in the sum of their instantaneous firing rates, $\hat{x}(t) = r_1(t) + r_2(t)$. If initially both neurons fired at 150 Hz, yielding a signal estimate $\hat{x} = 300$, then the sudden loss of one neuron will immediately degrade the signal estimate to $\hat{x} = 150$. More generally, in the absence of any corrective action, the signal estimate will be half the size it used to be.

There are two solutions to this problem. First, a downstream or read-out area could recover the correct signal estimate by doubling its decoder weight for the remaining neuron. In this scenario, the two-neuron circuit would remain oblivious to the loss of a neuron, and the burden of adjustment would be borne by downstream areas. Second, the firing rate of the remaining neuron could double, so that a downstream area would still obtain the correct signal estimate. In this scenario, the two-neuron circuit would correct the problem itself, and downstream areas would remain unaffected. While actual neural systems may make use of both possibilities, here we will study the second solution, and show that it has crucial advantages.

### The principle of optimal compensation

To quantify the ability of a system to compensate for neuron loss, we will assume that the representation performance can be measured with a simple cost-benefit trade-off. We formalize this trade-off in a loss function,

$$E = (\mathbf{x} - \hat{\mathbf{x}})^2 + \beta C(\mathbf{r}).$$  (2)

Here, the first term quantifies the signal representation error—the smaller the difference between the readout, $\hat{\mathbf{x}}$, and the actual signals, $\mathbf{x}$, the smaller the representation error. The second term

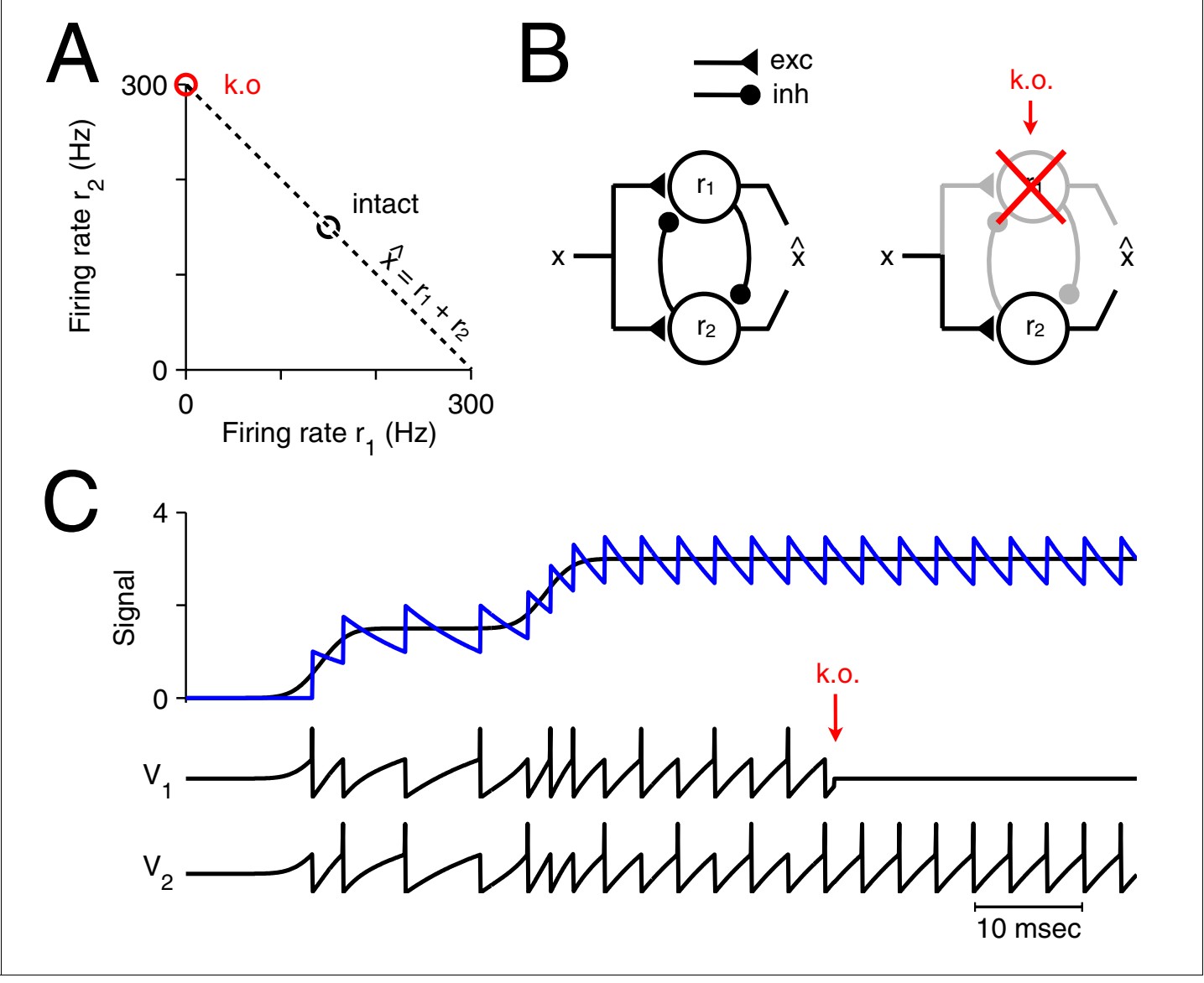

**Figure 1.** Optimal representation and optimal compensation in a two-neuron example. (**A**) A signal *x* can be represented using many different firing rate combinations (dashed line). Before cell loss, the combination that best shares the load between the neurons, requires that both cells are equally active (black circle). After the loss of neuron 1, the signal must be represented entirely by the remaining neuron, and so, its firing rate must double (red circle). We will refer to this change in firing rate as an optimal compensation for the lost neuron. (**B**) This optimal compensation can be implemented in a network of neurons that are coupled by mutual inhibition and driven by an excitatory input signal. When neuron 1 is knocked out, the inhibition disappears, allowing the firing rate of neuron 2 to increase. (**C**) Two spiking neurons, connected together as in B, represent a signal *x* (black line, top panel) by producing appropriate spike trains and voltage traces, $V_1$ and $V_2$ (lower panels). To read out the original signal, the spikes are replaced by decaying exponentials (a simple model of postsynaptic potentials), and summed to produce a readout, $\hat{x}$ (blue, top panel). After the loss of neuron 1 (red arrow), neuron 2 compensates by firing twice as many spikes.

quantifies the cost of the representation and is problem-specific and system-specific. Generally, we assume that the costs are small if the signal representation (1) is shared amongst all neurons and (2) uses as few spikes as possible. The parameter *β* determines the trade-off between this cost and the representation error. Quadratic loss functions such as *Equation 2* have been a mainstay of efficient coding theories for many years, for example, in stimulus reconstruction (*Rieke et al., 1997*) and in sparse coding (*Olshausen and Field, 1996*).

The minimum of our loss function indicates the set of firing rates that represent a given signal optimally. In the two-neuron example in *Figure 1A*, the initial minimum is reached if both neurons fire equally strong (assuming a cost of $C(\mathbf{r}) = r_1^2 + r_2^2$). If we kill neuron 1, then we can minimize the loss function under the constraint that $r_1 = 0$, and the minimum is now reached if neuron 2 doubles its firing rates (assuming a small trade-off parameter $\beta$). The induced change in the firing rate of neuron 2 preserves the signal representation, and therefore compensates for the loss of neuron 1. More generally, if the firing rates of a neural system represent a given signal optimally with respect to some loss function, then we define 'optimal compensation' as the change in firing rates necessary to remain at the minimum of that loss function.

Clearly, a system's ability to compensate for a lost neuron relies on some redundancy in the representation. If there are more neurons than signals, then different combinations of firing rates may represent the same signal (see *Figure 1A*, dashed line, for the two-neuron example). Up to a certain point (which we will study below), a neural system can always restore signal representation by adjusting its firing rates.

## Optimal compensation through instantaneous restoration of balance

Next, we investigate how this compensation can be implemented in a neural circuit. One possibility is that a circuit rewires through internal plasticity mechanisms in order to correct for the effect of the lost neurons. However, we find that plasticity is not necessary. Rather, neural networks can be wired up such that the dynamics of their firing rates rapidly evolves into the minimum of the loss function, *Equation 2* (*Hopfield, 1984*; *Dayan and Abbott, 2001*; *Rozell et al., 2008*). Such dynamics can be formulated not just for rate networks, but also for networks of integrate-and-fire neurons (*Boerlin et al., 2013*). The step towards spiking neurons will lead us to two crucial insights. First, we will link the optimal compensation principle to the balance of excitation and inhibition. Second, unlike networks of idealized rate units, networks of spiking neurons can only generate positive firing rates. We will show that this constraint alters the nature of compensation in non-trivial ways.

In order to move towards spiking networks, we assume that the instantaneous firing rates, $r_i(t)$, are equivalent to filtered spike trains, similar to the postsynaptic filtering of actual spike trains (see Materials and methods for details). Specifically, a spike fired by neuron $i$ contributes a discrete unit to its instantaneous firing rate, $r_i \to r_i + 1$, followed by an exponential decay. Next, we assume that each neuron will only fire a spike if the resulting change in the instantaneous firing rate reduces the loss function (*Equation 2*). From these two assumptions, the dynamics and connectivity of the network can be derived (*Boerlin et al., 2013*; *Bourdoukan et al., 2012*; *Barrett et al., 2013*). Even though neurons in these networks are ordinary integrate-and-fire neurons, the voltage of each neuron acquires an important functional interpretation, as it represents a transformation of the signal representation error, $\mathbf{x} - \hat{\mathbf{x}}$ (see Materials and methods).

If we apply this formalism to the two-neuron example in *Figure 1*, we obtain a network with two integrate-and-fire neurons driven by an excitatory input signal and coupled by mutual inhibition (*Figure 1B*). If we neglect the cost term, then the neuron's (subthreshold) voltages directly reflect the difference between the scalar input signal, $x(t)$, and its linear readout, $\hat{x}(t)$, so that $V_i = D(x - \hat{x})$ for $i = \{1, 2\}$ (*Figure 1C*). Initially, the excitatory input signal depolarizes the voltages of both neurons, which reflects an increase in the signal representation error. The neuron that reaches threshold first produces a spike and corrects the signal readout, $\hat{x}(t)$. In turn, the errors or voltages in both neurons are reset, due to the self-reset of the spiking neuron, and due to fast inhibition of the other neuron. This procedure repeats, so that both neurons work together, and take turns at producing spikes (*Figure 1C*). Now, when one neuron dies (*Figure 1C*, red arrow), the remaining neuron no longer receives inhibition from its partner neuron and it becomes much more strongly excited, spiking twice as often.

These ideas can be scaled up to larger networks of excitatory and inhibitory neurons, as illustrated in *Figure 2*. Here a homogeneous population of excitatory neurons, again modeled as leaky integrate-and-fire neurons (*Figure 2A*), is driven by a time-varying input signal, $x(t)$. Just as in the simplified two-neuron example, the excitatory neurons take turns at generating spikes, such that the linear readout of their spike trains, *Equation 1*, yields an estimate $\hat{x}(t)$ that minimizes the loss function from *Equation 2* (*Figure 2D*).

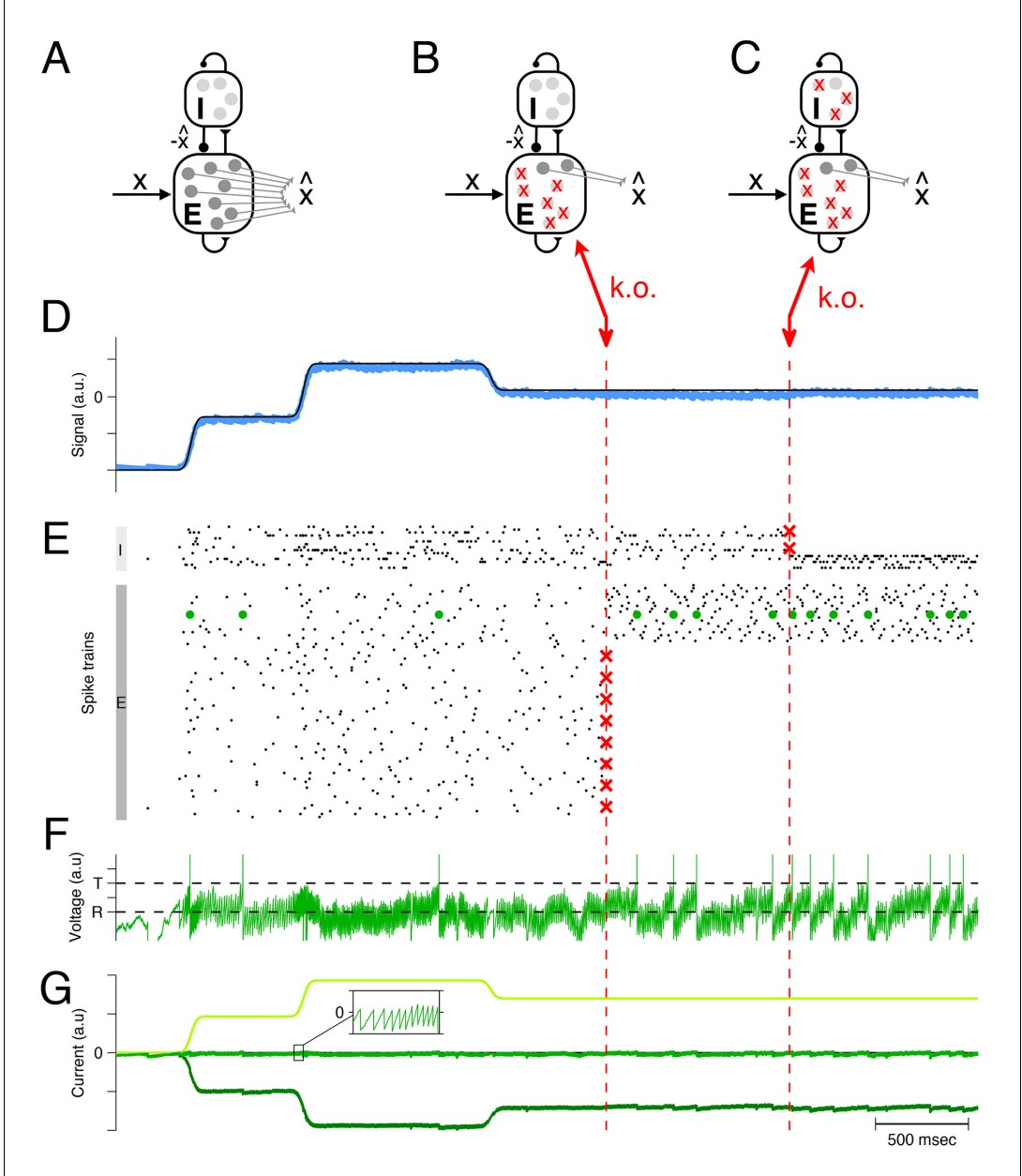

**Figure 2.** Optimal compensation in a larger spiking network with separate excitatory (E) and inhibitory (I) populations ($N = 80$ excitatory and $N = 20$ inhibitory neurons) (A) Schematic of network representing a scalar signal $x(t)$. The excitatory neurons receive an excitatory feedforward input, that reflects the input signal, $x(t)$, and a recurrent inhibitory input, that reflects the signal estimate, $\hat{x}(t)$. The signal estimate stems from the excitatory population and is simply re-routed through the inhibitory population. In turn, the voltages of the excitatory neurons are given by a transformation of the signal representation error, $V_i = D_i(x(t) - \hat{x}(t))$, assuming that $\beta = 0$. Since a neuron's voltage is bounded from above by the threshold, all excesses in signal representation errors are eliminated by spiking, and the signal estimate remains close to the signal, especially if the input signals are large compared to the threshold. Mechanistically, excitatory inputs here generate signal representation errors, and inhibitory inputs eliminate them. The prompt resolution of all errors is therefore identical to a precise balance of excitation and inhibition. (B) Schematic of a network with 75% of the excitatory neurons knocked out. (C) Schematic of a network with 75% of the inhibitory neurons knocked out also. (D) The network provides an accurate representation $\hat{x}(t)$ (blue line) of the time-varying signal $x(t)$ (black line). Whether part of the excitatory population is knocked out or part of the inhibitory population is knocked out, the representation remains intact, even though the readout weights do not change. (E) A raster plot of the spiking activity in the inhibitory and excitatory populations. In both populations, spike trains are quite irregular. Red crosses mark the portion of neurons that are knocked out. Whenever neurons are knocked out, the remaining neurons change their firing rates instantaneously and restore the signal representation. (F) Voltage of an example neuron (marked with green dots in E) with threshold $T$ and reset potential $R$. (G) The total excitatory (light green) and inhibitory (dark green) input currents to this example neuron are balanced, and the remaining fluctuations (inset) produce the membrane potential shown in F. This balance is maintained after the partial knock-outs in the E and I populations.

Mechanistically, the excitatory population receives an excitatory input signal, $x(t)$, that is matched by an equally strong inhibitory input. This inhibitory input is generated by a separate population of inhibitory interneurons, and is effectively equivalent to the (re-routed) signal estimate, $\hat{x}(t)$. As a result, the (subthreshold) voltage of each excitatory neuron is once again determined by the signal representation error, $x(t) - \hat{x}(t)$, and a neuron will only fire when this error exceeds its (voltage) threshold. By design, each spike improves the signal estimate, and decreases the voltages. The spike rasters of both populations are shown in *Figure 2E*.

Since the signals are excitatory and the signal estimates are inhibitory, the tight tracking of the signal by the signal estimate corresponds to a tight tracking of excitation by inhibition. The network is therefore an example of a balanced network in which inhibition and excitation track each other spike by spike (*Denève and Machens, 2016*). Importantly, the excitatory-inhibitory (EI) balance is a direct correlate of the signal representation accuracy (see Materials and methods for mathematical details).

Now, when some portion of the excitatory neural population is eliminated (*Figure 2B*), the inhibitory neurons receive less excitation, and in turn convey less inhibition to the remaining excitatory neurons. Consequently, the excitatory neurons increase their firing rates, and the levels of inhibition (or excitation) readjust automatically such that EI balance is restored (see *Figure 2G* for an example neuron). The restoration of balance guarantees that the signals continue to be represented accurately (*Figure 2D*). This compensation happens because each excitatory neuron seeks to keep the signal representation error in check, with or without the help of the other excitatory neurons. When one neuron dies, the remaining excitatory neurons automatically and rapidly assume the full burden of signal representation.

Similar principles hold when part of the inhibitory population is knocked out (*Figure 2C*). The inhibitory population here is constructed to faithfully represent its 'input signals', which are given by the postsynaptically filtered spike trains of the excitatory neurons. Consequently, the inhibitory population acts similarly to the excitatory population, and its response to neuron loss is almost identical. When inhibitory neurons are knocked out, the remaining inhibitory neurons increase their firing rates in order to restore the EI balance. Again, the compensation happens because each inhibitory neuron seeks to keep its signal representation error in check, independent of the other inhibitory neurons.

## The recovery boundary

Naturally, the recovery from neural loss is not unlimited. As we will show now, the resultant *recovery boundary* is marked by a breakdown in the balance of excitation and inhibition. In *Figure 2*, this recovery boundary coincides with a complete knock-out of either the excitatory or the inhibitory population. The nature of the recovery boundary becomes more complex, however, if a network tracks more than one signal.

In *Figure 3* we show an example network that represents two sinusoidal signals, $\mathbf{x}(t) = (x_1(t), x_2(t))$. The spikes of each neuron contribute to two readouts, and the decoding weights $\mathbf{D}_i$ of the neurons correspond to two-dimensional vectors (*Figure 3D*). The two signals and the corresponding signal estimates are shown in *Figure 3E*, and the spike trains of the excitatory population are shown in *Figure 3F*. (Here and in the following, we will focus on neuron loss in the excitatory population only, and replace the inhibitory population by direct inhibitory connections between excitatory neurons, see *Figure 3A–C*. This simplification does not affect the compensatory properties of the network, see Materials and methods.)

When part of the excitatory population is lost (*Figure 3B,D*), the remaining neurons compensate by adjusting their firing rates, and the signal representation remains intact (*Figure 3E,F*, first k.o.). However, when too many cells are eliminated from the network (*Figure 3C,D*), some portions of the signal can no longer be represented, no matter how the remaining neurons change their firing rates (*Figure 3E,F*, second k.o.). In this example, the recovery boundary occurs when all the neurons with negative-valued readout weights along the $x_1$-axis have been lost, so that the network can no longer represent the negative component of the first signal (*Figure 3E*, second k.o.). The representation of signal $x_2(t)$ is largely unaffected, so the deficits are only partial.

The recovery boundary is characterized by the network's inability to correct signal representation errors. Since the neurons' voltages correspond to transformations of these signal representation errors, the recovery boundary occurs when some of these voltages run out of bounds (*Figure 3G*, arrows). Such aberrations of the voltages are caused by an excess of either positive or negative

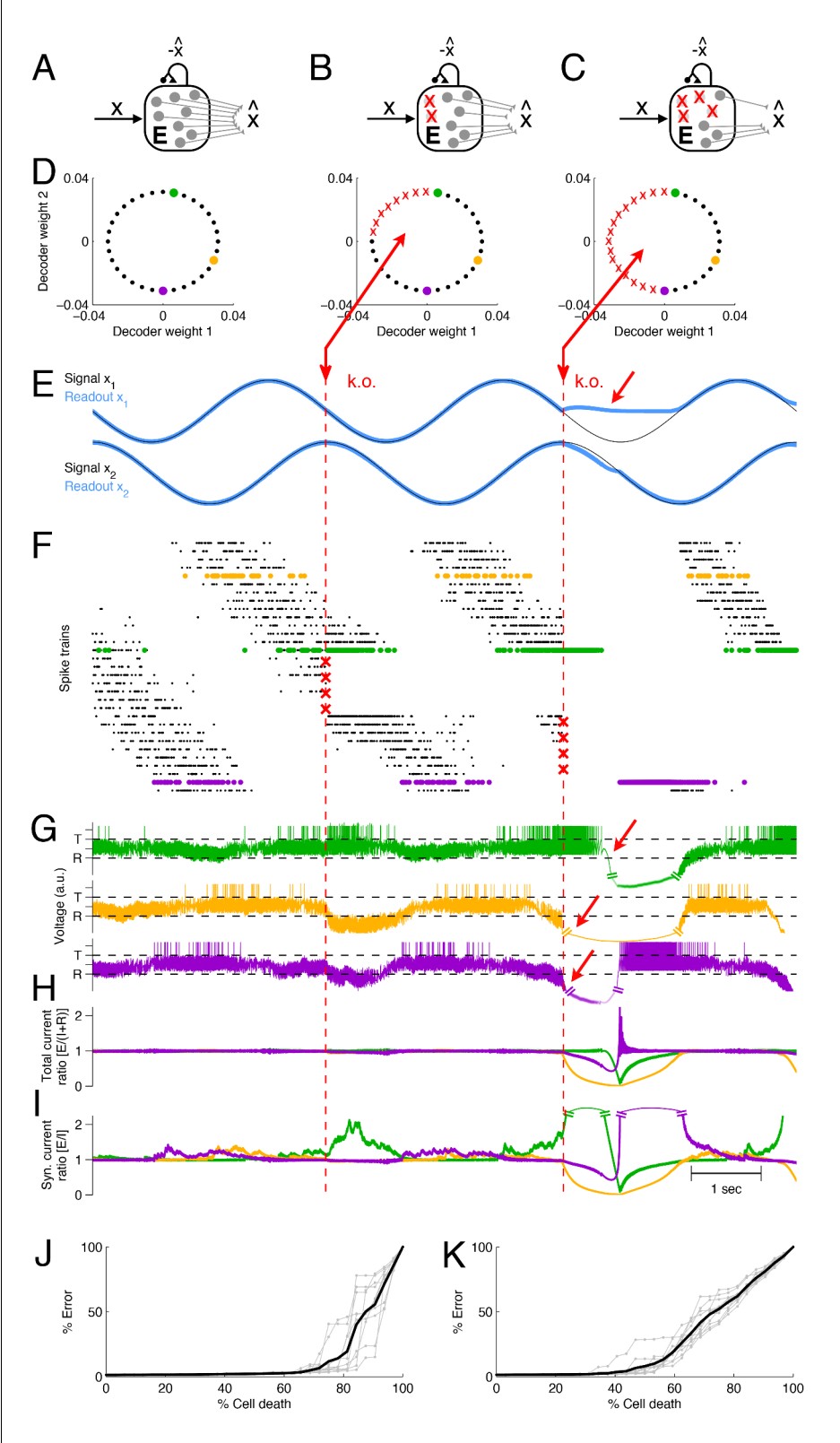

**Figure 3.** Optimal compensation and recovery boundary in a network with $N = 32$ excitatory neurons representing a two-dimensional signal $\mathbf{x}(t) = (x_1(t), x_2(t))$. (A–C) Schematic of network and knock-out schedule. (Inhibitory neurons not explicitly modeled for simplicity, see text.) (B) Neural knock-out that leads to optimal compensation. (C) Neural knock-out that pushes the system beyond the recovery boundary. (D) Readout or decoding weights of the neurons. Since the signal is two-dimensional, each neuron contributes to the readout with two decoding weights, shown here as black

*Figure 3 continued on next page*

*Figure 3 continued*

dots. Red crosses in the central and right panel indicate neurons that are knocked out at the respective time. (**E**) Signals and readouts. In this example, the two signals (black lines) consist of a sine and cosine. The network provides an accurate representation (blue lines) of the signals, even after 25% of the neurons have been knocked out (first dashed red line). When 50% of the neurons are knocked out (second dashed red line), the representation of the first signal fails temporarily (red arrow). (**F**) A raster plot of the spiking activity of all neurons. Neurons take turns in participating in the representation of the two signals (which trace out a circle in two dimensions), and individual spike trains are quite irregular. Red crosses mark the portion of neurons that are knocked out. Colored neurons (green, yellow, and purple dots) correspond to the neurons with the decoder weights shown in panel D. Note that the green and purple neurons undergo dramatic changes in firing rate after the first and second knock-outs, respectively, which reflects their attempts to restore the signal representation. (**G**) Voltages of the three example neurons (marked with green, yellow, and purple dots in D and F), with threshold $T$ and reset potential $R$. (**H**) Ratio of positive (E; excitation) and negative (I+R; inhibition and reset) currents in the three example neurons. After the first knock-out, the ratio remains around one, and the system remains balanced despite the loss of neurons. After the second knock-out, part of the signal can no longer be represented, and balance in the example neurons is (temporarily) lost. (**I**) Ratio of excitatory and inhibitory currents. Similar to H, except that a neuron's self-reset current is not included. (**J**) When neurons are knocked out at random, the network can withstand much larger percentages of cell loss. Individual grey traces show ten different, random knock-out schemes, and the black trace shows the average. (**K**) When neurons saturate, the overall ability of the network to compensate for neuron loss decreases.

currents. Indeed, the ratio of positive to negative currents becomes disrupted after the second k.o., precisely when $x_1(t)$ becomes negative (*Figure 3H*). Whereas a neuron can always 'eliminate' a positive, depolarizing current by spiking, it cannot eliminate any negative, hyperpolarizing currents, and these latter currents therefore reflect signal representation errors that are not eliminated within the network.

Here, positive currents are generated by excitatory inputs, whereas negative currents are generated by inhibitory inputs as well as the self-reset currents after an action potential. The self-reset currents become negligible in networks in which the number of neurons, $N$, is much larger than the number of input signals, $M$ (see *Figure 2*; see also Materials and methods). Even in *Figure 3* this is approximately the case, so that the ratio of positive to negative currents is similar to the ratio of excitatory to inhibitory synaptic inputs (*Figure 3I*). The recovery boundary is then signified by a dramatic breakdown in the balance of excitatory and inhibitory currents.

In this simple example, the recovery boundary emerged because neurons were knocked out in a systematic and ordered fashion (according to the features that they represented). Such a scenario may correspond to lesions or knock-outs in systems in which neural tuning is topologically organized. If the representational features of neurons are randomly interspersed, we find a different scenario. Indeed, when neurons are knocked out at random in our model, the network can withstand the loss of many more neurons. We find that signals are well represented until only a few neurons remain (*Figure 3J*).

At the recovery boundary, the few remaining neurons will shoulder the full representational load. This can easily become an unwieldy situation because the firing rates of the remaining neurons must become unrealistically large to compensate for all the lost neurons. In reality, neural activity will saturate at some point, in which case the recovery boundary occurs much earlier. For instance, we find that the example network tolerates up to 70–80% of random neuron loss if the maximum firing rate of a neuron is unbounded (*Figure 3J*), whereas it will only tolerate 40–50% of random neuron loss if the maximum firing rate is 80 Hz (*Figure 3K*).

## The influence of idle neurons on tuning curve shapes

So far we have considered what happens when neurons are lost permanently in a neural circuit, either because of their death or because of experimental interventions. We have shown that networks can compensate for this loss up to a recovery boundary. Independent of whether compensation is successful or not, it is always marked by changes in firing rates. We will now show how to quantify and predict these changes.

To do so, we first notice that neurons can also become inactive temporarily in a quite natural manner. In sensory systems, for instance, certain stimuli may inhibit neurons and silence them. This natural, temporary inactivity of neurons leads to similar, 'compensatory' effects on the population level in our networks. These effects will provide important clues to understanding how neurons' firing rates need to change whenever neurons become inactive, whether in a natural or non-natural context.

For simplicity, let us focus on constant input signals, $\mathbf{x}(t) = \mathbf{x}$, and study the *average* firing rates of the neurons, $\bar{\mathbf{r}}$. For constant inputs, the average firing rates must approach the minimum of the same loss function as in *Equation 2* so that,

$$\bar{\mathbf{r}}(\mathbf{x}) = \arg\min_{\bar{r}_i \geq 0, \forall i}\left[(\mathbf{x} - \hat{\mathbf{x}})^2 + \beta C(\bar{\mathbf{r}})\right] \tag{3}$$

where the signal estimate, $\hat{\mathbf{x}} = \sum_i \mathbf{D}_i \bar{r}_i$, is formed using the average firing rates. This minimization is performed under the constraint that firing rates must be positive, since the firing rates of spiking neurons are positive valued quantities, by definition. Mathematically, the corresponding optimization problem is known as 'quadratic programming' (see Materials and methods for more details). Traditionally, studies of population coding or efficient coding assume both positive and negative firing rates (*Dayan and Abbott, 2001*; *Olshausen and Field, 1996*; *Simoncelli and Olshausen, 2001*), and the restriction to positive firing rates is generally considered an implementational problem (*Rozell et al., 2008*). Surprisingly, however, we find that the constraint changes the nature of the solutions and provides fundamental insights into the shapes of tuning curves in our networks, as clarified below.

An example is shown in *Figure 4* where we focus on networks of increasing complexity that represent two signals. For simplicity, we keep the second signal constant ('fixed background' signal, $x_2 = r_B$) and study the firing rates of all neurons as a function of the first signal, $x_1 = x$. Solving *Equation 3* for a range of values, we obtain $N$ neural tuning curves, $\bar{\mathbf{r}}(x_1) = (r_1(x_1), r_2(x_1), \ldots)$ (*Figure 4*, second column). These tuning curves closely match those measured in simulations of the corresponding spiking networks (*Figure 4*, third and fourth column; see supplementary materials for mathematical details).

We observe that the positivity constraint produces non-linearities in neural tuning curves. We illustrate this using a two-neuron system with two opposite-valued readout weights for the first signal (*Figure 4A*, first column). At signal value $x = 0$, both neurons fire at equal rates so that the readout $\hat{x}$, *Equation 1*, correctly becomes $\hat{x} \propto (\bar{r}_1 - \bar{r}_2) = 0$. When we move to higher values of $x$, the firing rate of the first neuron, $\bar{r}_1$, increases linearly (*Figure 4A*, second column, orange line), and the firing rate of the second neuron, $\bar{r}_2$, decreases linearly (*Figure 4A*, second column, blue line), so that in each case, $\hat{x} \approx x$. Eventually, around $x = 0.4$, the firing rate of neuron two hits zero (*Figure 4A*, black arrow). At this point, neuron two has effectively disappeared from the network. Since it cannot decrease below zero, and because the estimated value $\hat{x}$ must keep growing with $x$, the firing rate of neuron one must grow at a faster rate. This causes a kink in its tuning curve (*Figure 4A*, black arrow). This kink in the tuning curve slope is an indirect form of optimal compensation, where the network is compensating for the temporary silencing of a neuron.

More generally, the firing rates of neurons are piecewise linear functions of the input signals. In *Figure 4B,C*, every time one of the neurons hits the zero firing rate lower bound, the tuning curve slopes of all the other neurons change. We furthermore note that the tuning curves also depend on the form of the cost terms, $C(\bar{\mathbf{r}})$. If there is no cost term, there are many equally favorable firing rate combinations that produce identical readouts (*Figure 1A*). The precise choice of a cost term determines which of these solutions is found by the network. We provide further geometric insights into the shape of the tuning curves obtained in our networks in *Figure 4—figure supplements 1–3* and in the Materials and methods.

## Tuning curves before and after neuronal silencing

We can now calculate how tuning curves change shape in our spiking network following neuronal loss. When a set of neurons are killed or silenced within a network, their firing rates are effectively set to zero. We can include this silencing of neurons in our loss function by simply clamping the respective neurons' firing rates to zero:

$$\bar{\mathbf{r}}(\mathbf{x}) = \arg\min_{\substack{\bar{r}_i \geq 0 \, if \, i \in X \\ \bar{r}_j = 0 \, if \, j \in Y}}\left[(\mathbf{x} - \hat{\mathbf{x}})^2 + \beta C(\bar{\mathbf{r}})\right] \tag{4}$$

where $X$ denotes the set of healthy neurons and $Y$ the set of dead (or silenced) neurons. This additional clamping constraint is the mathematical equivalent of killing neurons. In turn, we can study the

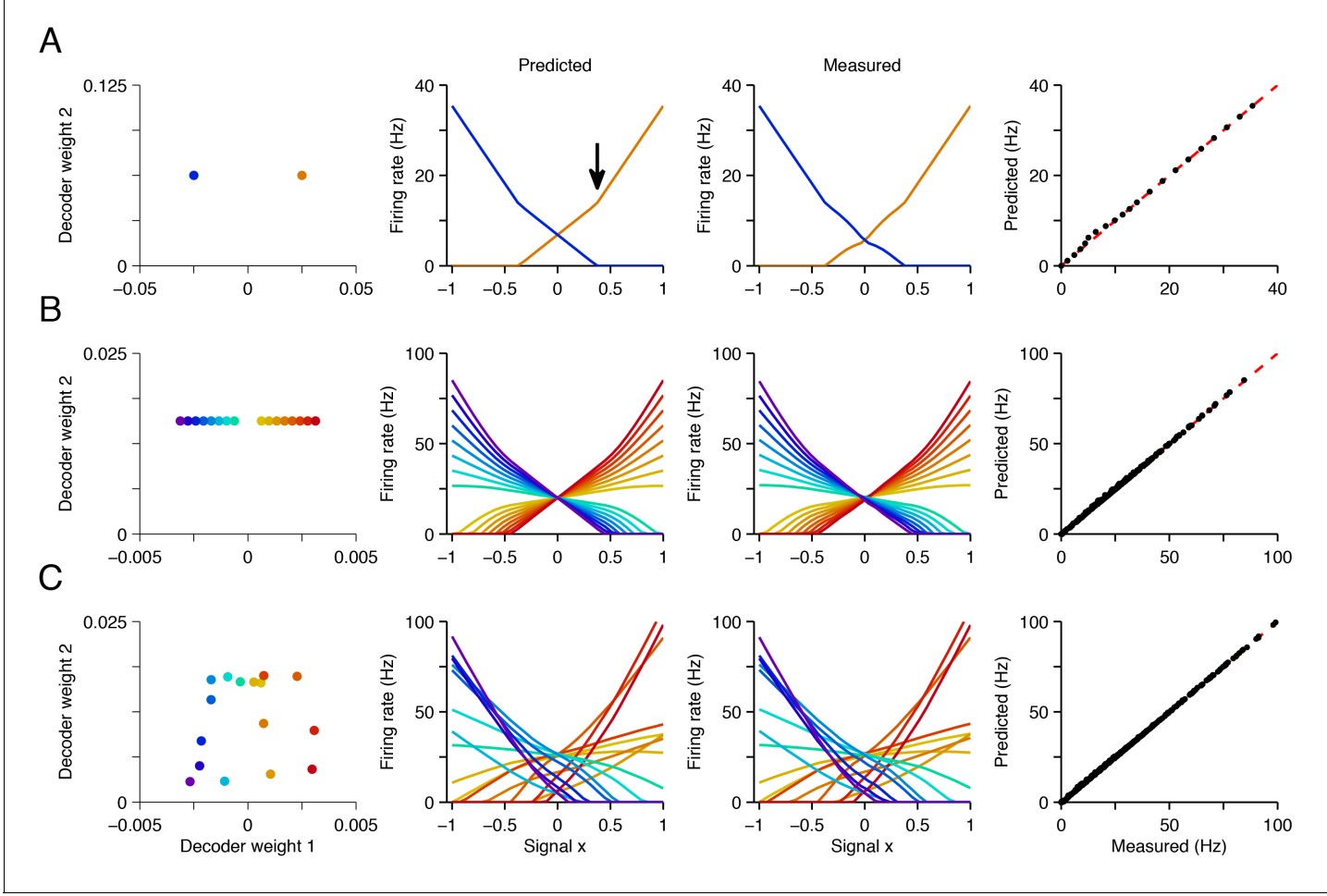

**Figure 4.** Explaining tuning curves in the spiking network with quadratic programming. (**A**) A network with $N = 2$ neurons. The first column shows the decoding weights for the two neurons. The first decoding weight determines the readout of the signal $x_1 = x$, whereas the second decoding weight helps to represent a constant background signal $x_2 = r_B$. The second column shows the tuning curves, i.e., firing rates as a function of $x$, predicted using quadratic programming. The third column shows the tuning curves measured during a 1 s simulation of the respective spiking network. The fourth column shows the match between the measured and predicted tuning curves. (**B**) Similar to A , but using a network of $N = 16$ neurons with inhomogeneous, regularly spaced decoding weights. The resulting tuning curves are regularly spaced. Neurons with small decoding weight magnitude have eccentric firing rate thresholds and shallow slopes; neurons with large decoding weight magnitudes have central firing rate thresholds and steep slopes. (**C**) Similar to B except that decoding weights and cost terms are irregularly spaced. This irregularity leads to inhomogeneity in the balanced network tuning curves, and in the quadratic programming prediction (see also *Figure 4—figure supplements 1–3*).

The following figure supplements are available for figure 4:

**Figure supplement 1.** The geometry of quadratic programming.

**Figure supplement 2.** A taxonomy of tuning curve shapes.

**Figure supplement 3.** Quadratic programming firing rate predictions compared to spiking network measurements.

tuning curves of neurons in networks with knocked-out neurons without having to simulate the corresponding spiking networks every single time.

As a first example, we revisit the network shown in *Figure 4C*, which represents a signal $x_1 = x$, and a constant 'background' signal $x_2 = r_B$. This network exhibits a complex mixture of tuning curves, i.e., firing rates as a function of the primary signal $x_1 = x$, with positive and negative slopes, and diverse threshold crossings (*Figure 5B*). When we knock out a subset of neurons (*Figure 5A*,

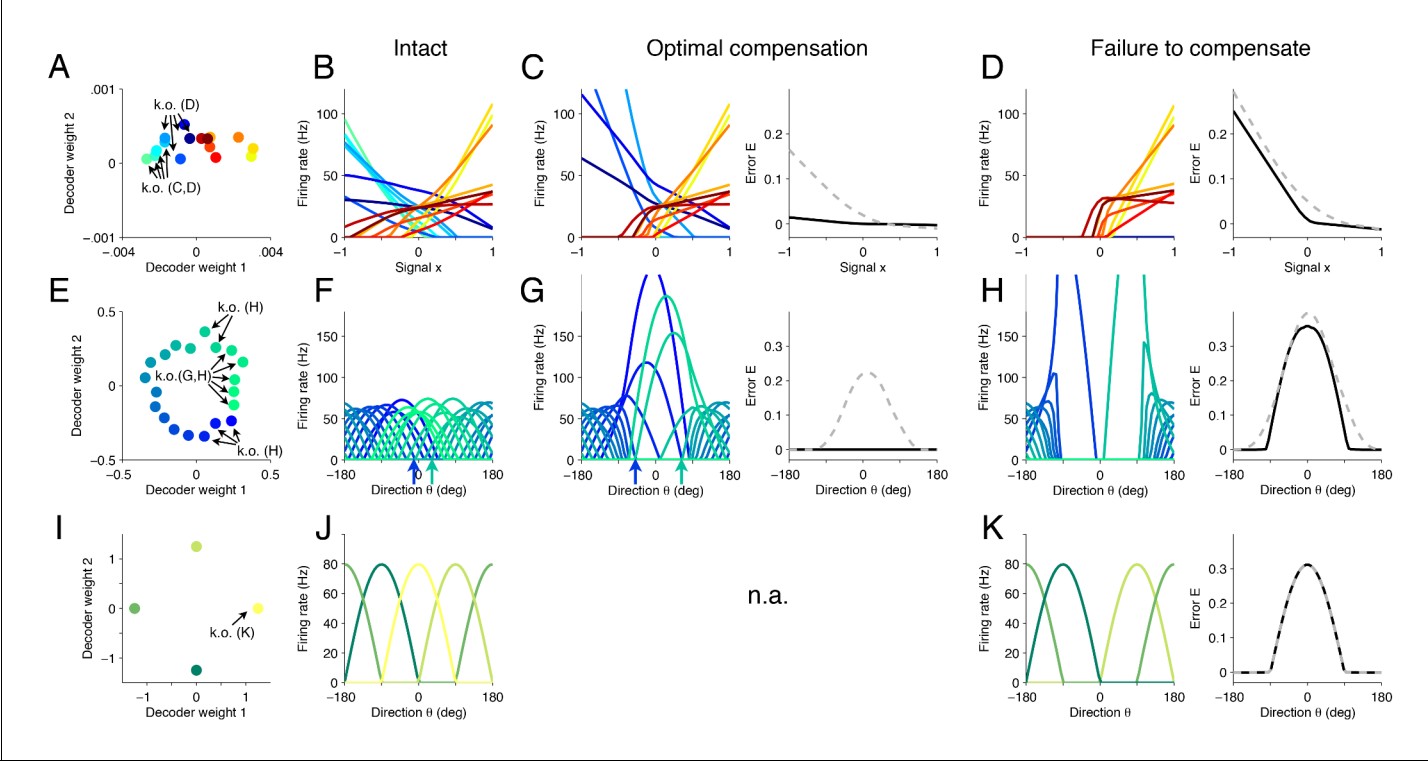

**Figure 5.** Changes in tuning curves following neuron loss and optimal compensation, as calculated with quadratic programming for three different networks. (A) Decoding weights for a network of $N = 16$ neurons, similar to *Figure 4C*. Arrows indicate the neurons that will be knocked out, either in panel (C) or (D). (B) The firing rates of all neurons given as a function of the first input signal $x = x_1$. Each line is the tuning curve of a single neuron, with either positive slope (yellow–red) or negative slope (blue-green). (C) Left, tuning curves after the loss of four negatively sloped neurons and after optimal compensation from the remaining neurons. Right, the signal representation continues to be accurate (black line). If the tuning curves of the remaining neurons do not undergo optimal compensation, the signal representation fails for negative values of $x$ (dashed grey line). (D) After the loss of the remaining negatively sloped neurons, the population is no longer capable of producing a readout $\hat{x}$ that can match negative values of the signal $x$, even with optimal compensation (black line). This happens because the recovery boundary has been reached – where signal representation fails. The changes in tuning curves that occur still seek to minimize the error, as can be seen when comparing the error incurred by a network whose tuning curves do not change (dashed gray line). (E) Decoding weights for a two-dimensional network model containing $N = 20$ neurons. Apart from the smaller number of neurons and the added heterogeneity, this network is equivalent to the network shown in *Figure 3*. (F) This system has bell-shaped tuning curves when firing rates are plotted as a function of a circular signal with direction $\theta$. (G) Left, when some of the neurons have been knocked out, the tuning curves of the remaining neurons change by shifting towards the preferred directions of the knocked out neurons. Right, this compensation preserves high signal representation quality (black line). In comparison, if the tuning curves do not change, the readout error increases substantially (dashed gray line). (H) Left, when more neurons are knocked-out (here, all neurons with positive decoding weights), the firing rates of the remaining neurons still shift towards the missing directions (compare highlighted spike trains in *Figure 3*). Right, despite the compensation, the network is no longer able to properly represent the signal in all directions. (I) Decoding weights for a two-dimensional network model containing $N = 4$ neurons. (J) The network exhibits four bell-shaped tuning curves. (K) Left, following the loss of a single neuron and optimal compensation, the tuning curves of the remaining neurons do not change. Right, after cell loss, the network incurs a strong error around the preferred direction of the knocked out neuron ($\theta = 0^o$), with optimal compensation (black line) or without optimal compensation (dashed gray line).

neurons labeled 'C'), we find that neurons with similar tuning to the knocked out neurons increase their firing rates and dissimilarly-tuned neurons decrease their firing rates, especially for negative values of the signal $x$ (*Figure 5C, left*). In this way, signal representation is preserved as much as possible (*Figure 5C*, right, black line). In comparison, a network that does not change its tuning curves after the cell loss has drastically worse representation error (*Figure 5C*, right, dashed line). Once all the neurons with positive tuning curve slopes are killed (*Figure 5A*, neurons labeled 'D'), we cross the recovery boundary, and even optimal compensation can no longer preserve the representation of negatively-valued signals (*Figure 5D*).

Next, we investigate these compensatory mechanisms in neural systems that have bell-shaped tuning curves (*Ben-Yishai et al., 1995*). Our framework captures these cases if we assume that a

network represents circular signals embedded in a two-dimensional space, $\mathbf{x} = (\cos(\theta), \sin(\theta))$, where $\theta$ is the direction of our signal. To do so, we construct a network in which the decoding weights of the neurons are spaced around a circle, with some irregularity added (*Figure 5E*). As a result, we obtain an irregular combination of bell-shaped tuning curves (*Figure 5F*), similar to those found in the primary visual or primary motor cortex, with each neuron having a different preferred direction and maximum firing rate as determined by the value of the decoding weights. We note that, apart from the smaller number of neurons and the greater irregularity of the decoding weights, this network is similar to the spiking network investigated in *Figure 3*.

If we now kill neurons within a specific range of preferred directions (*Figure 5E*), then the remaining neurons with the most similar directions increase their firing rates and shift their tuning curves towards the preferred directions of the missing neurons (*Figure 5G*, left). Furthermore, neurons further away from the knocked-out region will slightly skew their tuning curves by decreasing their firing rates for directions around zero (*Figure 5F,G*, arrows). Overall, the portion of space that is under-represented following cell loss becomes populated by neighboring neurons, which thereby counter-act the loss. In turn, the signal representation performance of the population is dramatically improved following optimal compensation compared to a system without compensatory mechanisms (*Figure 5G*, right). Once all neurons with positive read-out weights along the first axis are killed, the network can no longer compensate, and signal representation fails, despite the strong changes in firing rates of the neurons (*Figure 5H*). This latter scenario is equivalent to the one we observed in *Figure 3* after the second knock-out.

The ability of the system to compensate for the loss of neurons depends on the redundancy of its representation (although redundancy alone is not sufficient for compensation). If our network consists of only four neurons with equally spaced readout weights (*Figure 5I*), then it cannot compensate for the loss of neurons. Rather, we find that this model system crosses the recovery boundary following the loss of a single neuron, and so, optimal compensation does not produce any changes in the remaining neurons (*Figure 5K*, left). It occurs because the four neurons exactly partition the four quadrants of the signal space, and the remaining neurons cannot make any changes that would improve the signal representation.

## Comparison to experimental data

Does optimal compensation happen in real neural systems? Our framework makes strong predictions for how firing rates should change following neuronal knock-outs. Unfortunately, there is currently little data on how neural firing rates change directly after the loss of neurons in a given system. However, we found data from three systems—the oculomotor integrator in the hindbrain, the cricket cercal system, and the primary visual cortex—which allows us to make a first comparison of our predictions to reality. We emphasize that a strict test of our theory is left for future work.

The first system is the horizontal velocity-to-position integrator of the vertebrate oculomotor system, which is responsible for horizontal eye fixations, and which is comparable to the network model in *Figure 5A–D*. This system is usually considered to represent the signals that need to be sent to the two muscles (lateral and medial rectus) that control horizontal eye movements. In this case, the signal $x_1$ corresponds to the eye position, controlled by the difference in muscular activity, with zero representing the central eye position, positive values representing right-side eye positions and negative values representing left-side eye positions. The fixed background signal corresponds to the net sum of the two muscle activities, which we assume to remain constant for simplicity.

We find that the tuning curves of neurons measured in the right half of the oculomotor system (*Aksay et al., 2000*) (*Figure 6A*) are similar to the positively-sloped tuning curves in our network model calculated using *Equation 3* (*Figure 6B*). In both cases, neurons that encode right-side eye positions have positive slopes, and these follow a recruitment order, where neurons with shallow tuning curve slopes become active before neurons with steep tuning curve slopes (*Fuchs et al., 1988*; *Aksay et al., 2000*; *Pastor and Gonzalez-Forero, 2003*) (*Figure 6A,B* inset). Similar observations hold for the left half of the oculomotor system, which has negatively-sloped tuning curves (data not shown).

Now, when the left half of the oculomotor system is inactivated using lidocaine and muscimol injections (*Aksay et al., 2007*), the system can still represent right-side eye positions, but is unable to represent left-side eye positions (*Figure 6C*). In our network model, this inactivation corresponds to eliminating all negatively-sloped tuning curves (*Figure 5D*). The optimal compensation ensures

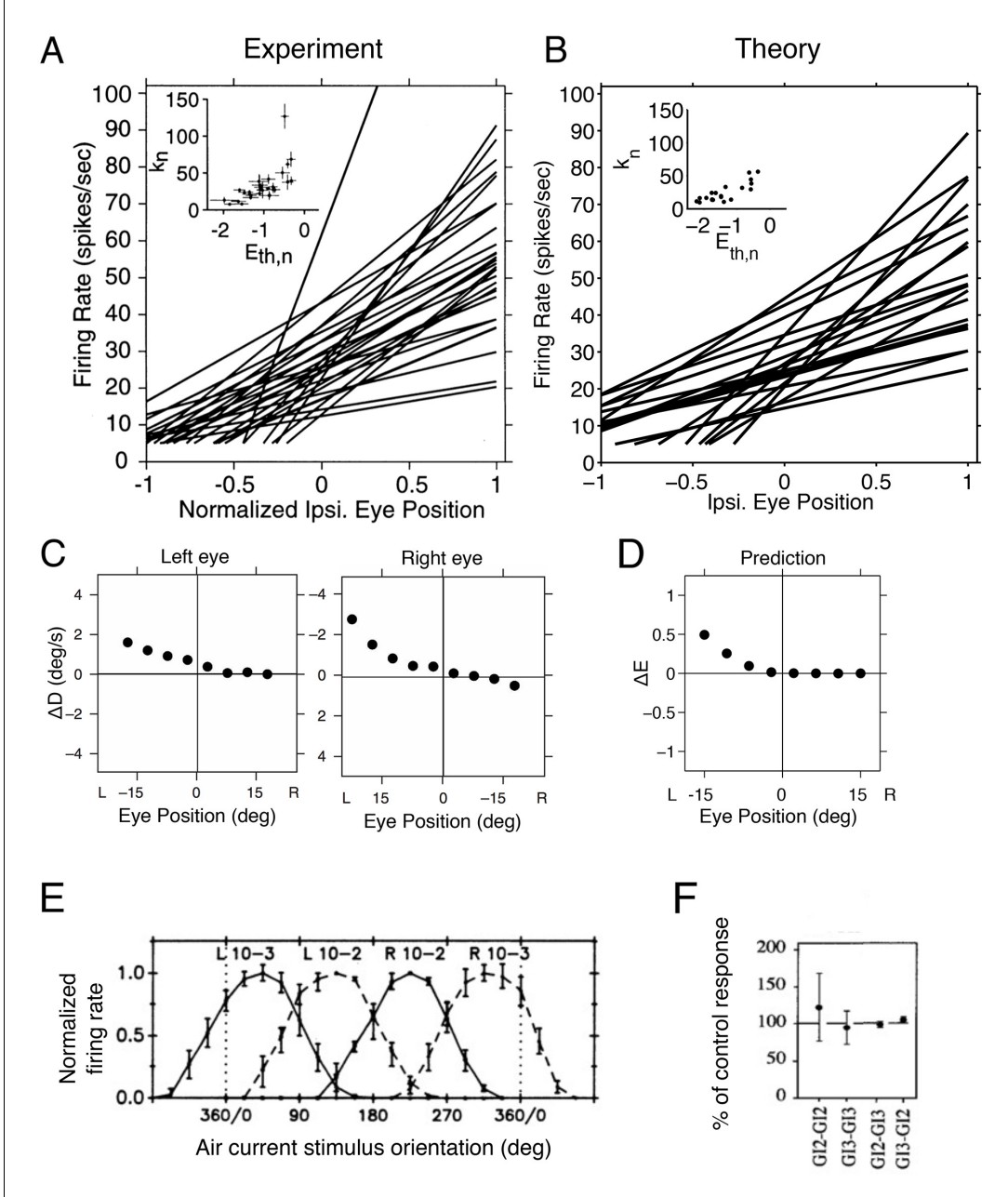

**Figure 6.** Tuning curves and inactivation experiments in the oculomotor integrator of the goldfish, and in the cricket cercal system. (**A**) Tuning curve measurements from *right-side* oculomotor neurons (Area 1 of goldfish). Firing rate measurements above 5 Hz are fit with a straight line $f_n = k_n(x - E_{th,n})$, where $f_n$ is the firing rate of the $n^{th}$ neuron, $E_{th,n}$ is the firing rate threshold, $k_n$ is the firing rate slope and $x$ is the eye position. As the eye position increases, from left to right, a recruitment order is observed, where neuron slopes increase as the firing rate threshold increases (inset).
*Figure 6A* has been adapted with permission from The American Physiological Society (© copyright The American Physiological Society, 2000. All Rights Reserved).

   (**B**) Tuning curves from our network model. We use the same parameters as in previous figures (*Figure 5A*), and fit the threshold-linear model to the simulated tuning curves (*Figure 5B*) using the same procedure as in the experiments from *Aksay et al. (2000)*. As in the data, a recruitment order is observed with slopes increasing as the firing threshold increases (inset). (**C**) Eye position drift measurements after the pharmacological inactivation of left side neurons in goldfish. Inactivation was performed using lidocaine and muscimol injections. Here $\Delta D = D_{\text{after}} - D_{\text{before}}$, where $D_{\text{before}}$ is the average drift in eye position before pharmacological inactivation and is the average drift in eye position after pharmacological inactivation. Averages are calculated across goldfish.
*Figure 6C* has been adapted with permission from McMillan Publisher Ltd: Nature Neuroscience (© copyright McMillan Publisher Ltd: Nature Neuroscience, 2007. All Rights Reserved).
*Figure 6 continued on next page*

*Figure 6 continued*

(D) Eye position representation error of the model network following optimal compensation. Here $\Delta E = E_C - E_I$, where is the representation error of the intact system and $E_C$ is the representation error following optimal compensation. These representation errors are calculated using the loss function from *Equation 2* . (E) Firing rate recordings from the cricket cercal system in response to air currents from different directions. Each neuron has a preference for a different wind direction. Compare with *Figure 5J*.
*Figure 6E* has been adapted with permission from The American Physiological Society (© copyright The American Physiological Society, 1991. All Rights Reserved).

(F) Measured change of cockroach cercal system tuning curves following the ablation of another cercal system neuron. In all cases, there is no significant change in firing rate. This is consistent with the lack of compensation in *Figure 5K*. The notation GI2-GI3 denotes that Giant Interneuron 2 is ablated and the change in Giant Interneuron 3 is measured. The firing rate after ablation is given as a percentage of the firing rate before ablation.
*Figure 6F* has been adapted with permission from The American Physiological Society (© copyright The American Physiological Society, 1997. All Rights Reserved).

that the right part of the signal is still correctly represented, but the left part can no longer be represented due to the lack of negatively tuned neurons (*Figure 5D* and *Figure 6D*). Hence, these measurements are consistent with our optimal compensation and recovery boundary predictions.

The second system is a small sensory system called the cricket (or cockroach) cercal system, which represents wind velocity (*Theunissen and Miller, 1991*), and which is comparable to the network model in *Figure 5I–K*. In the case of the cercal system, $\theta$ is the wind-direction. The tuning curves we obtain are similar to those measured, with each neuron having a different, preferred wind direction (*Theunissen and Miller, 1991*), see *Figure 6E*. The similarity between measured tuning curves (*Figure 6E*) and predicted tuning curves (*Figure 5J*) suggests that we can interpret cercal system neurons to be optimal for representing wind direction using equally spaced readout weights.

When one neuron in the cercal system is killed, the remaining neurons do not change their firing rates (*Libersat and Mizrahi, 1996*; *Mizrahi and Libersat, 1997*), see *Figure 6F*. This is analogous to our predictions (*Figure 5K*). Indeed, our model of the cercal system has no redundancy (see Materials and methods for more details). The cercal system in all its simplicity is therefore a nice example of a system that exists on the 'edge' of the recovery boundary.

## A high-dimensional example: optimal compensation in V1

The relatively simple models we have described so far are useful for understanding the principle of optimal compensation, but they are unlikely to capture the complexity of large neural populations in the brain. Even though the network model in *Figure 5E–G* resembles common network models for the primary visual or primary motor cortex, it does not capture the high-dimensional nature of representations found in these areas. To investigate the compensatory mechanisms in more complex networks, we make a final generalization to systems that can represent high-dimensional signals, and for concreteness we focus on the (primary) visual cortex, which is our third test system.

The visual cortex is generally thought to construct representations of images consisting of many pixel values (*Figure 7A*). The tuning of V1 simple cells, for instance, can largely be accounted for by assuming that neural firing rates provide a sparse code of natural images (*Olshausen and Field, 1996*; *Simoncelli and Olshausen, 2001*). In accordance with this theory, we choose a model that represents image patches (size $12 \times 12$, corresponding to $M = 144$ dimensions), and we use decoding weights that are optimized for natural images (see Materials and methods). Neurons in this model are tuned to both the orientation and polarity of edge-like images (*Figure 7D*), where the polarity is either a bright edge with dark flanks, or the opposite polarity – a dark edge with bright flanks. Orientation tuning emerges because natural images typically contain edges at many different orientations, and a sparse code captures these natural statistics (*Olshausen and Field, 1996*; *Simoncelli and Olshausen, 2001*). Polarity tuning emerges as a natural consequence of the positivity constraint, because a neuron with a positive firing rate cannot represent edges at two opposing polarities. Similar polarity tuning has been obtained before, but with an additional constraint that decoding weights be strictly positive (*Hoyer, 2003*, *2004*).

As before, we compute the firing rates of our neural population by solving *Equation 3*, which provides a convenient approximation of the underlying spiking network. We then silence all neurons with a vertical orientation preference (*Figure 7E*) and calculate the resulting changes in firing rates

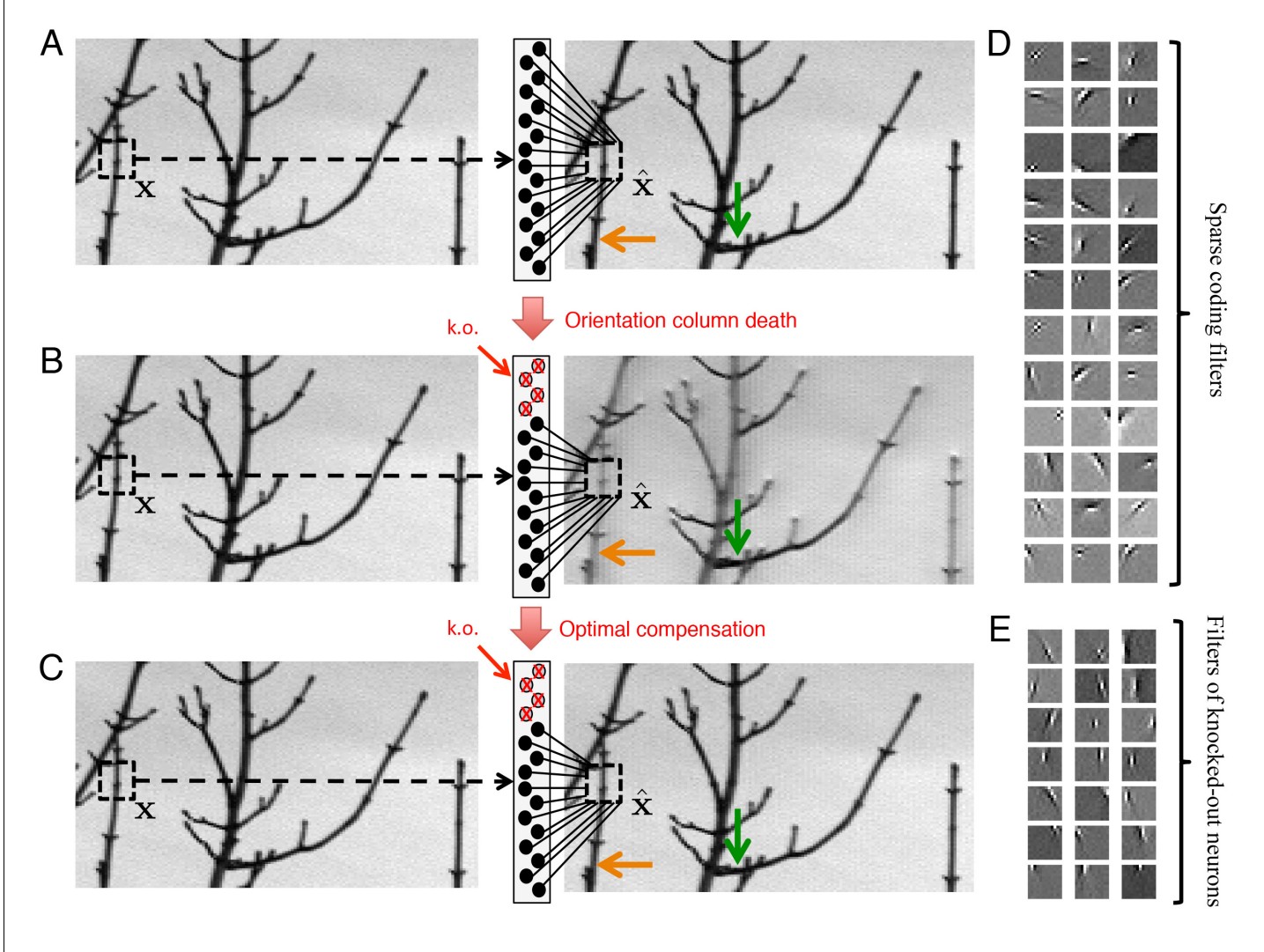

**Figure 7.** Optimal compensation for orientation column cell loss in a positive sparse coding model of the visual cortex. (A) Schematic of a neural population (middle) providing a representation (right) of a natural image (left). This image representation is formed when neurons respond to an image patch **x** with a sparse representation and an output $\hat{\mathbf{x}}$. Here, patches are overlapped to remove aliasing artifacts. (B) Following the loss of neurons that represent vertical orientations, the image representation degrades substantially without optimal compensation, especially for image segments that contain vertical lines (orange arrow), and less so for image segments that contain horizontal lines (green arrow). (C) Following the optimal compensation, the image representation is recovered. (D) A selection of efficient sparse decoding weights, illustrated as image patches. Each image patch represents the decoding weights for a single neuron. In total, there are 288 neurons in the population. (E) A selection of vertically oriented decoding weights whose neurons are selected for simulated loss. All neurons with preferred vertical orientations at $67.5^o$, $90^o$ and $112.5^o$ and at the opposite polarity, $-67.5^o$, $-90^o$ and $-112.5^o$ are silenced.

in the network. Without optimal compensation, (i.e., without any changes in firing rates), we find that the silencing of an orientation column damages image representation, especially for image components that contain edges parallel to the preferred orientations of the dead neurons (*Figure 7B*, orange arrow). When the population implements optimal compensation, the firing rates of many neurons change, and the image representation is recovered (*Figure 7C*).

To illustrate the nature of the network compensation, we study how the tuning of neurons changes when part of the population is lost. This focus will allow us to compare our predictions to experimental data. The tuning curves of all cells in response to the directions of various gratings are shown in *Figure 8A*. If we knock out a (small) subset of the cells, e.g., 50% of all neurons with preferred directions around $\theta = 0°$, then the firing rates of several of the remaining neurons increase at

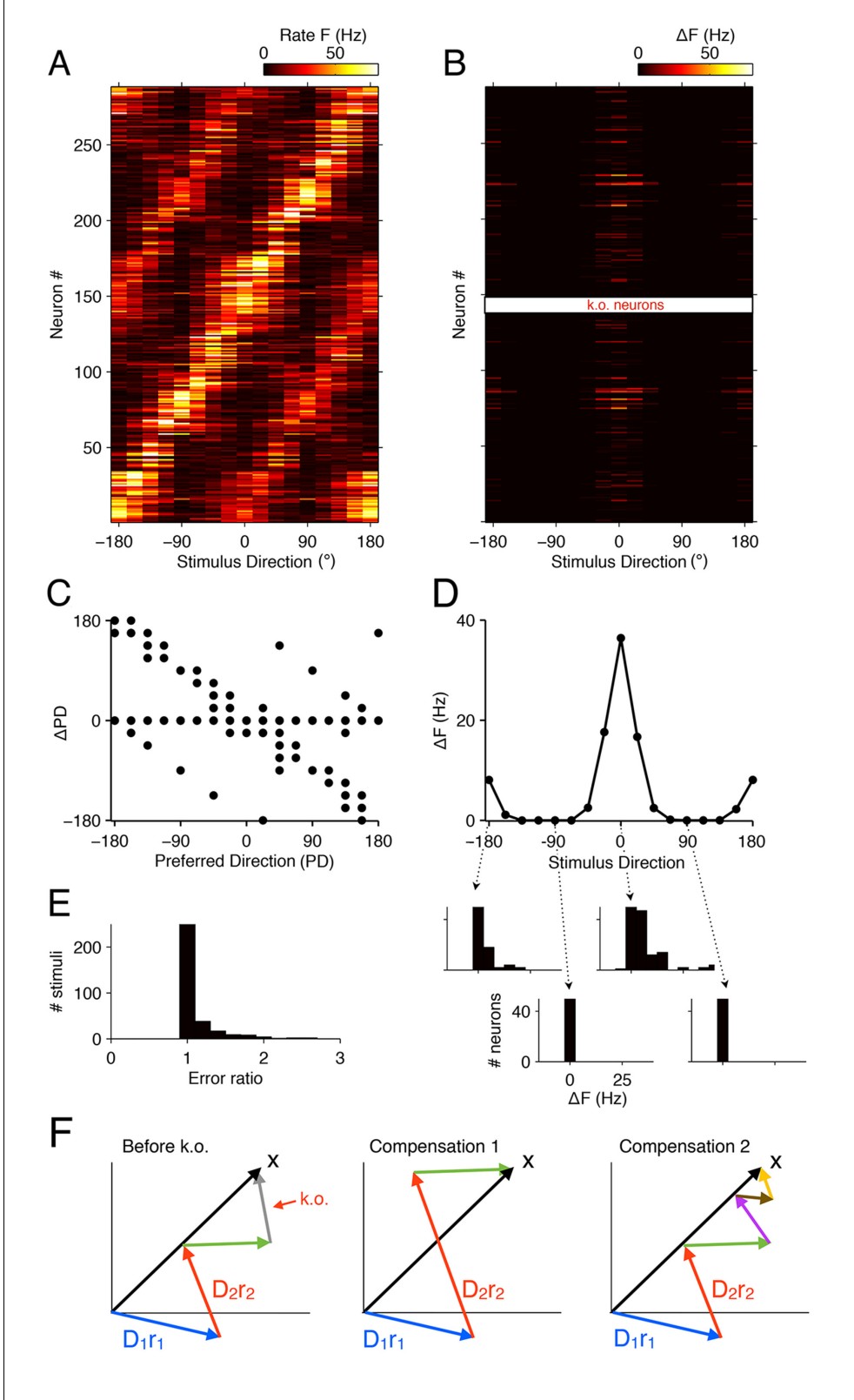

**Figure 8.** Tuning curves and signatures of optimal compensation in the V1 model. (**A**) The tuning curves (firing rates as a function of stimulus direction) for all neurons in the model ($N = 288$). (**B**) Changes in firing rates ($\Delta F$), as a function of stimulus direction, for all neurons in the model, when 50% of neurons with a preferred direction at $0°$ degrees are knocked out ($N = 11$). Several cells change their firing close to the preferred ($0°$) or anti-preferred ($-180°$ or $180°$) direction of the k.o. cells. (**C**) The change in preferred direction, $\Delta$PD, following optimal compensation is given for each neuron as a

*Figure 8 continued on next page*

*Figure 8 continued*

function of its preferred direction before cell loss. Many cells shift their preferred direction towards the preferred direction of the knocked out cells. (**D**) The average change in firing rate due to optimal compensation ($\Delta\overline{F}$) is calculated at each stimulus direction, where the average is taken across the neural population in B. There is a substantial increase in firing rates close to the preferred directions of the knocked out cells. The histograms on the bottom show the firing rate changes in the population as a function of four different stimulus directions. Stimuli with horizontal orientations ($0°$ or $\pm180°$) lead to firing rate changes, but stimuli with vertical orientations ($\pm90°$) do not. (**E**) Ratio of reconstruction errors (no compensation vs. optimal compensation) across a large range of Gabor stimuli. For most stimuli, no compensation is necessary, but for a few stimuli (mostly those showing edges with horizontal orientations) the errors are substantially reduced if the network compensates. (**F**) Schematic of compensation in high-dimensional spaces. In the intact state (left), a stimulus (black arrow) is represented through the firing of four neurons (colored arrows, representing the decoding vectors of the neurons, weighted by their firing rates). If the grey neuron is knocked out, one possibility to restore the representation may be to modify the firing rates of the remaining neurons (center). However, in higher-dimensional spaces, the three remaining arrows will not be sufficient to reach any stimulus position, so that this possibility is usually ruled out. Rather, additional neurons will need to be recruited (right) in order to restore the stimulus representation. These neurons may have contributed only marginally (or not at all) to the initial stimulus representation, because their contribution was too costly. In the absence of the grey neuron, however, they have become affordable. If no combination of neurons allows the system to reach the stimulus position, then the recovery boundary has been crossed.

the preferred direction of the silenced cells (*Figure 8B,D*). As a consequence, the preferred directions of many cells shift toward the preferred direction of the silenced cells (*Figure 8C*). Unlike the simple example in *Figure 5G*, however, these shifts occur in neurons throughout the population, and not just in neurons with neighboring directions. For the affected stimuli, the shifts decrease the representation errors compared to a system without compensation (*Figure 8E*).

The more complicated pattern of firing rate changes occurs for two reasons. First, neurons represent more than the direction of a grating—they represent a $12 \times 12$ image patch, i.e., a vector in a high-dimensional space (*Figure 8F*). Such a vector will generally be represented by the firing of many different neurons, each of which contributes with its decoding vector, $\mathbf{D}_i$, weighted by its firing rate, $r_i$. However, these neurons will rarely point exactly in the right (stimulus) direction, and only their combined effort allows them to represent the stimulus. When some of the neurons contributing to this stimulus representation are knocked out, the network will recruit from the pool of all neurons whose decoding vectors are non-orthogonal to the stimulus vector. In turn, each of these remaining neurons gets assigned a new firing rate such that the cost term in *Equation 2* is minimized. Second, high-dimensional spaces literally provide a lot of space. Whereas stacking 20 neurons in a two-dimensional space (as in *Figure 5E,F*) immediately causes crowding, with neighboring neurons sharing a substantial overlap of information, stacking e.g. 1000 neurons in a 100-dimensional space causes no such thing. Rather, neurons can remain almost independent, while sharing only small amounts of information. As a consequence, not a single pair of the naturally trained decoding weights in our model will point in similar directions. In other words, when we knock out a set of neurons, there are no similarly tuned neurons that could change their firing rates in order to restore the representation. Rather, the compensation for a single lost neuron will require the concerted effort of several other neurons, and many of the required changes can be small and subtle. However, all of these changes will seek to restore the lost information, and cells will shift their tuning towards that of the lost neurons.

As a consequence of this additional complexity, we find that firing rates at the anti-preferred direction also increase (*Figure 8B,D*). A key reason for this behavior is that many neurons are strongly tuned to orientation, but only weakly tuned to direction, so that eliminating neurons with $0°$ degree direction preference also eliminates some of the representational power at the anti-preferred direction.

Finally, although our model is simply a (positive) sparse coding model, we note that most of its predictions are consistent with experiments in the cat visual cortex, in which sites with specific orientation or direction tuning are silenced using GABA (*Crook et al., 1996*, *1997*, *1998*) (*Figure 9* and *Figure 9—figure supplement 1*), while recording the firing rates of nearby neurons. Neurons that escape silencing in the cat visual cortex also shift their tuning curves towards the preferred directions of the silenced neurons (*Figure 9A*). More specifically, three types of changes were found: (1) Neurons whose preferred direction was different to that of the silenced neurons (PD>22.5°) will increase their firing rates at the knocked-out direction and broaden their tuning (see *Figure 9A*, top row for an example). (2) Neurons whose preferred direction was opposite to the silenced neurons will

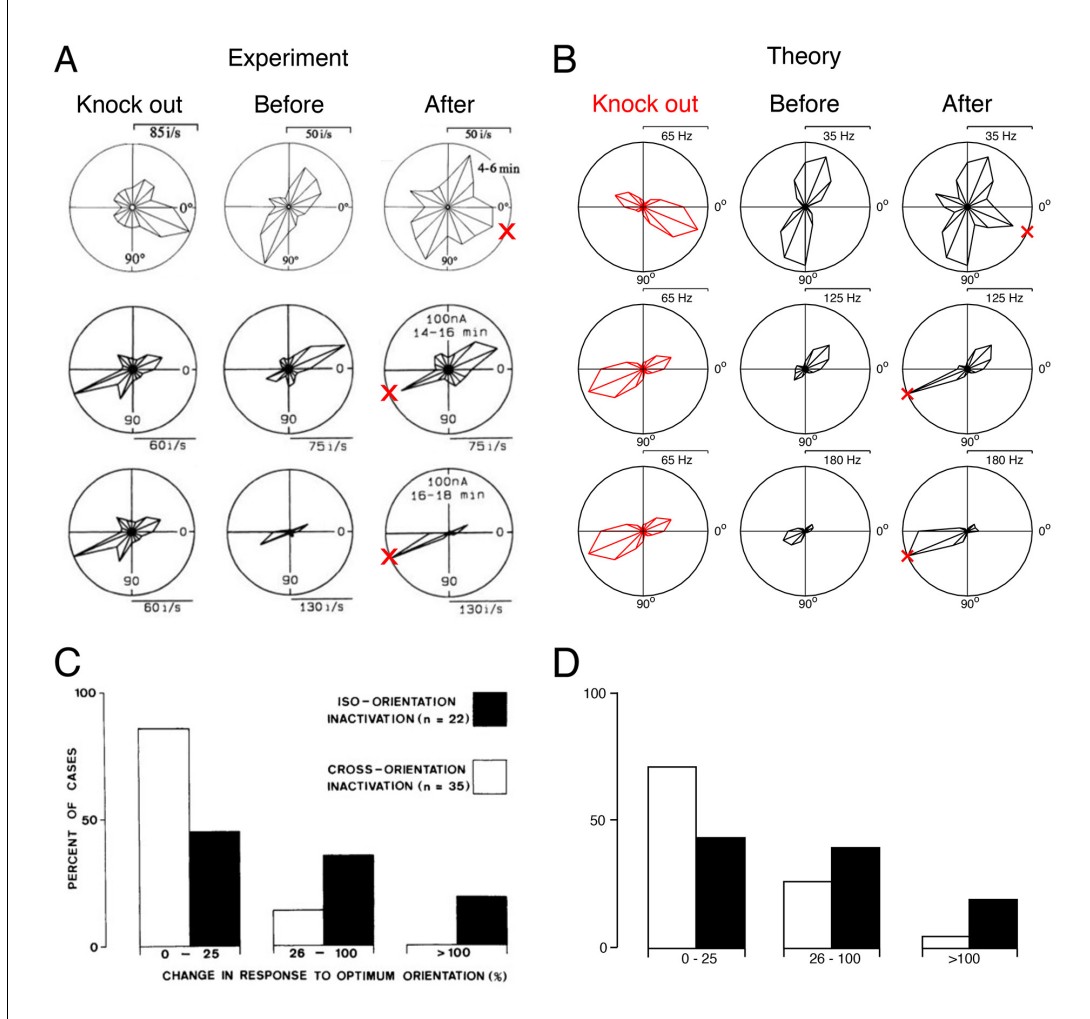

**Figure 9.** Tuning curves and inactivation experiments: Comparison between recordings from visual cortex and the V1 model. (**A**) Recordings of cat visual cortex neurons both before (middle column) and after (right column) the GABA-ergic knock out of neighboring neurons (left column). The tuning curves of all neurons are shown in polar coordinates. The firing rates of all neurons increase in the direction of the preferred direction of the silenced neurons (red cross). Each row illustrates the response of a different test neuron to silencing. Examples are selected for ease of comparison with the theory. Knock-out measurements were obtained using multi-unit recording, and single-unit recordings were obtained for neurons that were not subject to silencing.
*Figure 9A* has been adapted with permission from The American Physiological Society (© copyright The American Physiological Society, 1996. All Rights Reserved).
*Figure 9A* has been adapted with permission from Society of Neuroscience (© copyright Society of Neuroscience, 1992. All Rights Reserved).
   (**B**) A set of similarly tuned neurons are selected for the artificial knock out in our V1 model. The tuning curve of a test neuron is shown before the selected neurons are knocked out (middle column) and after optimal compensation (right column). Each row illustrates the response of a different test neuron to neuron loss. Following the optimal compensation, the neurons remaining after cell loss shift their tuning curves towards the preferred direction of the knocked out neurons (indicated by a red cross). (**C**) Histogram of changes in the firing rate at preferred stimulus orientations following GABA-ergic silencing. Firing rate change for neurons with tuning preferences that are similar to the recorded neurons (iso-orientation inactivation) are counted separately to changes in neurons with different tuning preferences (cross-orientation inactivation).
*Figure 9C* is reproduced with permission from Society of Neuroscience (© copyright Society of Neuroscience, 1992. All Rights Reserved).
   (**D**) Histogram firing rate changes in the V1 model, same format as C. In B and D we knock out 50% of neurons with preferred directions across a range of $50^o$ (see also *Figure 9—figure supplement 1*).

The following figure supplements are available for figure 9:

**Figure supplement 1.** Histograms of experimental responses to neuron silencing in V1, compared to theoretical predictions using a range of different parameters.

*Figure 9 continued on next page*

*Figure 9 continued*

**Figure supplement 2.** Optimal compensation in V1 models with different degrees of redundancy or over-completeness.

increase their firing rates towards the knocked out direction (see *Figure 9A*, middle row for an example). (3) Neurons whose preferred direction was similar to the silenced neurons (PD<22.5°) will either increase their firing to the knocked out direction (see *Figure 9A*, bottom row) or decrease their firing to the knocked out direction (data not shown).

In our sparse coding model, we find increases in firing rates to the knocked-out directions in all three types of neurons (with different direction preference, opposite direction preference, or similar direction preference), see *Figure 9B*. We do not observe a decrease in firing to the knocked-out directions. While this mismatch likely points towards the limits of our simplified model (which models a relatively small set of simple cells) in comparison with the data (which examines both simple and complex cells in both V1 and V2), it does not violate the optimal compensation principle. Due to the complexity of interactions in a high-dimensional space, a decrease in firing of a few neurons to the missing directions is not a priori ruled out in our model, as long as there is a net increase to the missing directions across the whole population. Indeed, when increasing the redundancy in our V1 model (e.g. by moving from $N = 288$ neurons to $N = 500$ neurons while keeping the dimensionality of the stimulus fixed), we find that several neurons start decreasing their firing rates in response to the knocked-out directions, even though the average firing rate changes in the population remain positive (data not shown).

We emphasize that there are only a few free parameters in our model. Since the decoding weights are learnt from natural images using the (positive) sparse coding model, we are left with the dimensionality of the input space, the number of neurons chosen, and the number of neurons knocked out. The qualitative results are largely independent of these parameters. For instance, the precise proportion of neurons knocked out in the experimental studies (*Crook and Eysel, 1992*) is not known. However, for a range of reasonable parameter choices (50–75% knock out of all neurons with a preferred direction), our predicted tuning curve changes are consistent with the recorded population response to neuron silencing (*Crook and Eysel, 1992*) (*Figure 9—figure supplement 1B*). Furthermore, if we increase the redundancy of the system by scaling up the number of neurons, but not the number of stimulus dimensions, we shift the recovery boundary to a larger fraction of knocked-out neurons, but we do not change the nature of the compensatory response (*Figure 9—figure supplement 2A,B*). At high degrees of redundancy (or so-called 'over-completeness'), quantitive fluctuations in tuning curve responses are averaged out, indicating that optimal compensation becomes invariant to over-completeness (*Figure 9—figure supplement 2C,D*). Given the few assumptions that enter our model, we believe that the broad qualitative agreement between our theory and the data is quite compelling.

## Discussion

To our knowledge, optimal compensation for neuron loss has not been proposed before. Usually, cell loss or neuron silencing is assumed to be a wholly destructive action, and the immediate neural response is assumed to be pathological, rather than corrective. Synaptic plasticity is typically given credit for the recovery of neural function. For example, synaptic compensation has been proposed as a mechanism for memory recovery following synaptic deletion (*Horn et al., 1996*), and optimal adaptation following perturbations of sensory stimuli (*Wainwright, 1999*) and motor targets (*Shadmehr et al., 2010*; *Braun et al., 2009*; *Kording et al., 2007*) has also been observed, but on a slow time scale consistent with synaptic plasticity. In this work, we have explored the properties of an instantaneous, optimal compensation, that can be implemented without synaptic plasticity, on a much faster time scale, through balanced network dynamics.

Our work is built upon a connection between two separate theories: the theory of balanced networks, which is widely regarded to be the standard model of cortical dynamics (*van Vreeswijk and Sompolinsky, 1996*, *1998*; *Amit and Brunel, 1997*; *Renart et al., 2010*; *Denève and Machens, 2016*) and the theory of efficient coding, which is arguably our most influential theory of neural computation (*Barlow, 1961*; *Salinas, 2006*; *Olshausen and Field, 1996*; *Bell and Sejnowski, 1997*;

*Greene et al., 2009*). This connection relies on the following two derivations: (1) We derive a tightly balanced spiking network from a quadratic loss function, following the recent work of (*Boerlin et al., 2013*; *Boerlin and Denève, 2011*) by focusing on the part of their networks that generate a spike-based representation of the information. (2) We show that the firing rates in these spiking networks also obey a quadratic loss function, albeit with a positivity constraint on the firing rate (*Barrett et al., 2013*). This constrained minimization problem, called quadratic programming, provides a novel link between spiking networks and firing rate calculations. While the importance of positivity constraints has been noted in other contexts (*Lee and Seung, 1999*; *Hoyer, 2003*, *2004*; *Salinas, 2006*), here we show its dramatic consequences in shaping tuning curves, which had not been appreciated previously. In turn, we obtain a single normative explanation for the polarity tuning of simple cells (*Figure 7D*), tuning curves in the oculomotor system (*Figure 6A,B*), and tuning curves in the cricket cercal system (*Figure 6E*), as well as the mechanisms underlying the generation of these tuning curves, and the response of these systems to neuron loss.

Several alternative network models have been proposed that minimize similar loss functions as in our work (*Dayan and Abbott, 2001*; *Rozell et al., 2008*; *Hu et al., 2012*; *Charles et al., 2012*; *Druckmann and Chklovskii, 2012*). However, in all these models, neurons produce positive and negative firing rates (*Dayan and Abbott, 2001*; *Rozell et al., 2008*; *Charles et al., 2012*; *Druckmann and Chklovskii, 2012*), or positive and negative valued spikes (*Hu et al., 2012*). The compensatory response of these systems will be radically different, because oppositely tuned neurons can compensate for each other by increasing their firing rates and changing sign, which is impossible in our networks. Similar reasoning holds for any efficient coding theories that assume positive and negative firing rates.

Whether general spiking networks support some type of compensation will depend on the specifics of their connectivity (see Materials and methods). For instance, spiking networks that learn to efficiently represent natural images could potentially compensate for neuron loss (*Zylberberg et al., 2011*; *King et al., 2013*). Furthermore, networks designed to generate EI balance through strong recurrent dynamics will automatically restore balance when neurons are lost (*van Vreeswijk and Sompolinsky, 1996*, *1998*; *Amit and Brunel, 1997*; *Renart et al., 2010*). In turn, networks whose representations and computations are based on such randomly balanced networks may be able to compensate for neuron loss as well (*Vogels et al., 2011*; *Hennequin et al., 2014*), as has been shown for neural integrators (*Lim and Goldman, 2013*). However, the optimality or the speed of the compensation is likely to increase with the tightness of the balancing. In turn, the tightness of EI balance is essentially determined by synaptic connectivity, whether through experience-dependent learning of the synaptic connections (*Haas et al., 2006*; *Vogels et al., 2011*; *Bourdoukan et al., 2012*; *Luz and Shamir, 2012*), through scaling laws for random connectivities (*van Vreeswijk and Sompolinsky, 1996*, *1998*; *Amit and Brunel, 1997*; *Renart et al., 2010*), or by design, via the decoding weights of linear readouts, as in our case (*Boerlin et al., 2013*; *Barrett et al., 2013*). Future work may resolve the commonalities and differences of these approaches.

The principle of optimal compensation is obviously an idealization, and any putative compensatory mechanism of an actual neural system may be more limited. That said, we have catalogued a series of experiments in which pharmacologically induced changes in neural activities and in behavior can be explained in terms of optimal compensation. These experiments were not originally conducted to test any specific compensatory mechanisms, and so, the results of each individual experiment were explained by separate, alternative mechanisms (*Aksay et al., 2007*; *Gonçalves et al., 2014*; *Libersat and Mizrahi, 1996*; *Mizrahi and Libersat, 1997*; *Crook et al., 1996*, *1997*, *1998*). The advantage of our optimal compensation theory is that it provides a simple, unifying explanation for all these experiments. Whether it is the correct explanation can only be answered through further experimental research.

To guide such research, we can use our theory to make a number of important predictions about the impact of neural damage on neural circuits. First, we can predict how tuning curves change shape to compensate for neuron loss. Specifically, neurons throughout the network will, on average, increase their firing rates to the signals that specifically activated the dead neurons. This happens because the remaining neurons automatically seek to carry the informational load of the knocked out neurons (which is equivalent to maintaining a balance of excitation and inhibition). This is a strong prediction of our theory, and as such, an observation inconsistent with this prediction would invalidate our theory. There have been very few experiments that measure neural tuning before and

after neuron silencing, but in the visual cortex, where this has been done, our predictions are consistent with experimental observations (*Figure 9*).

Our second prediction is that optimal compensation is extremely fast—faster than the time scale of neural spiking. This speed is possible because optimal compensation is supported by the tight balance of excitation and inhibition, which responds rapidly to neuron loss—just as balanced networks can respond rapidly to changes in inputs (*Renart et al., 2010*; *van Vreeswijk and Sompolinsky, 1998*, *1996*). We are not aware of any experiments that have explicitly tested the speed of firing rate changes following neuron silencing. In the pharmacological silencing of direction-selective cells in the visual cortex, the time scales of the reagents are too slow to out-rule the possibility that there is some synaptic plasticity (*Crook et al., 1996*, *1997*, *1998*). Nonetheless, these experiments are consistent with our prediction, because the changes observed in tuning curve shape are at least as fast, if not faster than the speed of pharmacological silencing. Ideally, these predictions could be tested using neuronal ablations or optogenetic silencing.

Finally, we predict that all neural systems have a cell loss recovery boundary, beyond which neural function disintegrates. Existing measurements from the oculomotor system (*Aksay et al., 2007*) seem to be consistent with this prediction (*Figure 6*). We predict that this recovery boundary coincides with a disruption in the balance of excitation and inhibition. This has not been explored experimentally, although the disruption of balance has recently been implicated in a range of neural disorders such as epilepsy (*Bromfield, 2006*) and schizophrenia (*Yizhar et al., 2011*). Anecdotally, there have been many unreported experiments where neural ablation has failed to cause a measurable behavioral effect. Our theory suggests that such 'failed' lesion experiments may be far more interesting than previously thought, and that the boundary between a measurable and unnoticeable behavioral effect deserves specific attention. Indeed, the properties of a recovery boundary may also shed light on the progression of neurodegenerative diseases—especially those that are characterized by a period of asymptomatic cell loss, followed by a transition to a disabled symptomatic state, as in Alzheimer's disease and stroke (*Leary and Saver, 2003*). We speculate that these transitions occur at the recovery boundary of the diseased system. If this were the case, then an accumulation of asymptomatic damage, through aging for example, or through acute conditions such as silent stroke, will increase the susceptibility of the brain to symptomatic damage, by moving it closer to the recovery boundary.

These predictions, and more broadly, the principle of optimal compensation that we have developed here, promise to be useful across a number of areas. First, as a neuroscience tool, our work provides a framework for the interpretation of experimental manipulations such as pharmacological silencing (*Aksay et al., 2007*), lesion studies (*Keck et al., 2008*) and optogenetic perturbations (*Fenno et al., 2011*; *Li et al., 2016*). Second, in the study of neural computation, optimal compensation may be a useful guiding principle, because plausible models of neural computation should be designed specifically to withstand the type of damage that the brain can withstand. Finally, our work may provide new perspectives on how neurodegenerative diseases impact behavior through neural networks, by generalizing the theory of efficient coding from the intact brain state to the damaged brain state (*Morrison and Hof, 1997*; *Bredesen et al., 2006*).

## Materials and methods

We have described the properties of optimal compensation, and given a variety of examples of optimal compensation across a range of systems. Here, we present further technical details. First, we describe how we tune spiking networks to represent signals optimally, including both networks that obey Dale's law and those that do not. Next, we explain quadratic programming with an analytically tractable example. We then specify our choice of parameters for each figure. For the high-dimensional sparse coding example, we describe how we calculate sparse coding receptive fields and direction tuning curves. Finally, we provide additional details on the knock-out calculations, and we prove that our spiking model is tightly balanced and performs optimal compensation. The Matlab code that we used to generate all the figures in this paper is published online (http://github.com/machenslab/spikes).

## Derivation of network model using spikes

In this section, we derive the connectivity and dynamics of a network that can optimally compensate for neuron loss. For simplicity, we will for now ignore the constraint that neurons need to be either excitatory or inhibitory, which will be revisited in the next section. We consider a network of $N$ leaky integrate-and-fire neurons receiving time-varying inputs $\mathbf{x}(t) = (x_1(t), \ldots, x_j(t), \ldots, x_M(t))$, where $M$ is the dimension of the input and $x_j(t)$ is the $j^{th}$ input signal. In response to this input, the network produces spike trains $\mathbf{s}(t) = (s_1(t), \ldots, s_i(t), \ldots, s_N(t))$, where $s_i(t) = \sum_l \delta(t - t_l^i)$ is the spike train of the $i^{th}$ neuron and $\{t_l^i\}$ are the spike times of that neuron. Here, we describe the general formulation of our framework for arbitrary networks with $N \geq M$.

A neuron fires a spike whenever its membrane potential exceeds a spiking threshold. We can write this as $V_i > T_i$, where $V_i$ is the membrane potential of neuron $i$ and $T_i$ is the spiking threshold. The dynamics of the membrane potentials are given by:

$$\frac{dV_i}{dt} = -\lambda V_i + \sum_{j=1}^{M} F_{ij} g(x_j) + \sum_{k=1}^{N} \Omega_{ik} s_k + \sigma_V \eta_i, \tag{5}$$

where $\Omega_{ik}$ is the connection strength from neuron $k$ to neuron $i$, $F_{ij}$ is the connection strength from input $j$ to neuron $i$, $g(x_j)$ is a function (or functional) applied to the input $x_j$, $\lambda$ is the neuron leak and $\sigma_V$ is the standard deviation of intrinsic neural noise, represented by a white noise term $\eta_i$ (**Knight, 1972**; **Dayan and Abbott, 2001**). For brevity, we will not explicitly indicate the time-dependency of variables. That said, we note that the input signals, $x_j(t)$, the voltages, $V_i(t)$, and the spike trains, $s_k(t)$, are all time-dependent quantities, whereas the thresholds, $T_i$, the leak, $\lambda$, and the connection strengths, $\Omega_{ik}$ and $F_{ij}$, are all constants. The feedforward term, $g(x_j)$, is a placeholder for any feedforward input into the networks, but also for slower recurrent synaptic inputs that are not explicitly modeled here, such as the slow synapses in **Boerlin et al. (2013)**. (Indeed, including slower recurrent dynamics does not affect the fast compensatory response of the network.) When a neuron spikes, its membrane potential is reset to the value $V = R_i$. For ease of presentation, we include this reset in the self-connections $\Omega_{ii}$, i.e., we assume that the self-connections are negative, $\Omega_{ii} < 0$, so that the reset is given by $R_i = T_i + \Omega_{ii}$.

We assume that this network provides a representation of the input signal $\mathbf{x}$ using a simple linear decoder, rewritten from **Equation 1**,

$$\hat{\mathbf{x}} = \sum_{k=1}^{N} \mathbf{D}_k r_k, \tag{6}$$

where $\mathbf{D}_k$ is the fixed contribution of neuron $k$ to the signal, and $r_k$ is the instantaneous firing rate of neuron $k$. We generally refer to $\mathbf{D}_k$ as the vector of readout or decoding weights. The instantaneous firing rate, $r_k$, is a time-dependent quantity that we obtain by filtering the spike train with an exponential filter:

$$r_k(t) = \int_0^\infty e^{-t'/\tau} s_k(t - t') dt', \tag{7}$$

where $\tau$ is the time-scale of the filtering. This firing rate definition is particularly informative because it has the form of a simple model of a postsynaptic potential, which is a biologically important quantity. Note that the units of this firing rate are given by $\tau$, so we must multiply by $\tau^{-1}$Hz to obtain units of Hz.

Our goal is to tune all the parameters of this network so that it produces appropriate spike trains at appropriate times to provide an accurate representation of the input $\mathbf{x}$, both before and after cell loss (**Boerlin et al., 2013**; **Bourdoukan et al., 2012**; **Barrett et al., 2013**). We measure the accuracy of the representation with the loss function, rewritten from **Equation 2**,

$$E = (\mathbf{x} - \hat{\mathbf{x}})^2 + \beta C(\mathbf{r}), \tag{8}$$

with $(\cdot)^2$ denoting an inner product, and $C(\mathbf{r}) = \sum_k r_k^2$. We then require our network to obey the

following rule: at a given time point $t$, each neuron only fires a spike whenever a spike reduces the loss function,

$$E(\text{no spike}) > E(\text{with spike}) \tag{9}$$

Then, since a spike changes the rate by $r_i \rightarrow r_i + 1$, and hence the signal estimate by $\hat{\mathbf{x}} \rightarrow \hat{\mathbf{x}} + \mathbf{D}_i$, we can restate this spiking rule for the $i$-th neuron as:

$$(\mathbf{x} - \hat{\mathbf{x}})^2 + \beta \sum_{k=1}^{N} r_k^2 \;>\; \left(\mathbf{x} - \left(\hat{\mathbf{x}} + \mathbf{D}_i\right)\right)^2 + \beta \sum_{k=1}^{N} (r_k + \delta_{ik})^2, \tag{10}$$

where $\delta_{ik}$ is the Kronecker delta. By expanding the right-hand-side, and canceling equal terms, this spiking rule can be rewritten as

$$\mathbf{D}_i^\top (\mathbf{x} - \hat{\mathbf{x}}) - \beta r_i > \frac{\mathbf{D}_i^\top \mathbf{D}_i + \beta}{2}. \tag{11}$$

This equation describes a rule under which neurons fire to produce spikes that reduce the loss function. Since a neuron $i$ spikes whenever its voltage $V_i$ exceeds its threshold $T_i$, we can interpret the left-hand-side of this spiking condition (*Equation 11*) as the membrane potential of the $i^{th}$ neuron:

$$V_i \equiv \mathbf{D}_i^\top (\mathbf{x} - \hat{\mathbf{x}}) - \beta r_i, \tag{12}$$

and the right-hand-side as the spiking threshold for that neuron:

$$T_i \equiv \frac{\mathbf{D}_i^\top \mathbf{D}_i + \beta}{2}. \tag{13}$$

We can identify the connectivity and the parameters that produce optimal coding spike trains by calculating the derivative of the membrane potential (as interpreted in *Equation 12*) and matching the result to the dynamical equations of our integrate-and-fire network (*Equation 5*):

$$\frac{dV_i}{dt} = \mathbf{D}_i^\top \left(\frac{d\mathbf{x}}{dt} - \frac{d\hat{\mathbf{x}}}{dt}\right) - \beta \frac{dr_i}{dt}. \tag{14}$$

From *Equation 6*, we obtain $d\hat{\mathbf{x}}/dt = \sum_k \mathbf{D}_k dr_k/dt$ and from *Equation 7* we obtain $dr_k/dt = -r_k/\tau + s_k$, which yields a simple differential equation for the readout:

$$\frac{d\hat{\mathbf{x}}}{dt} = -\frac{1}{\tau}\hat{\mathbf{x}} + \sum_{k=1}^{N} \mathbf{D}_k s_k. \tag{15}$$

By inserting these expressions into *Equation 14* we obtain:

$$\frac{dV_i}{dt} = \mathbf{D}_i^\top \frac{d\mathbf{x}}{dt} + \frac{1}{\tau}\mathbf{D}_i^\top \hat{\mathbf{x}} - \sum_{k=1}^{N}\left(\mathbf{D}_i^\top \mathbf{D}_k s_k\right) + \frac{1}{\tau}\beta r_i - \beta s_i \tag{16}$$

Finally, using the voltage definition from *Equation 12* to write $\mathbf{D}_i^\top \hat{\mathbf{x}} = -V_i + \mathbf{D}_i^\top \mathbf{x} - \beta r_i$ we can replace the second term on the right hand side and obtain:

$$\frac{dV_i}{dt} = -\frac{1}{\tau}V_i + \mathbf{D}_i^\top \left(\frac{d\mathbf{x}}{dt} + \frac{\mathbf{x}}{\tau}\right) - \sum_{k=1}^{N}\left(\mathbf{D}_i^\top \mathbf{D}_k + \beta \delta_{ik}\right)s_k. \tag{17}$$

This equation describes the voltage dynamics of a neuron that produces spikes to represent signal $\mathbf{x}$. If we now compare this equation with our original integrate-and-fire network, *Equation 5*, we see that these are equivalent if

$$\Omega_{ik} \equiv -\left(\mathbf{D}_i^\top \mathbf{D}_k + \beta \delta_{ik}\right), \tag{18}$$

$$F_{ij} \equiv [\mathbf{D}_i]_j, \tag{19}$$

$$g(x_j) \equiv \frac{dx_j}{dt} + \frac{x_j}{\tau}, \tag{20}$$

$$\lambda \equiv \frac{1}{\tau}, \tag{21}$$

$$T_i \equiv \frac{\mathbf{D}_i^\top \mathbf{D}_i + \beta}{2}. \tag{22}$$

Here, the notation $[\mathbf{z}]_j$ refers to the $j$-th element of the vector $\mathbf{z}$. A network of integrate-and-fire neurons with these parameters and connection strengths can produce spike trains that represent the signal $\mathbf{x}$ to a high degree of accuracy. Elements of this calculation have been produced before (*Boerlin et al., 2013*), but are reproduced here for the sake of completeness. Also, it has been shown that this connectivity can be learned using simple spike timing-dependent plasticity rules (W. Brendel, R. Bourdoukan, P. Vertechi, et al, unpublished observations; *Bourdoukan et al. (2012)*), so extensive fine-tuning is not required to obtain these spiking networks. We note that the input into the network consists of a combination of the original signal, $x_j$, and its derivative, $dx_j/dt$. This can be interpreted in several ways. First, we can think of the original signal, $x_j(t)$, as a filtered version of the actual input, so that $dx_j/dt = -\lambda x_j + c_j(t)$, where $c_j(t)$ is the actual input (and note that $c_j = \lambda x_j + dx_j/dt$). Second, we can think of this as a biophysical computation, in the sense that the neuron receives two types of input, the original signal input, $x_j(t)$, and its derivative, $dx_j/dt$. The latter could be computed through a simple circuit that combines direct excitatory signal inputs with delayed inhibitory signal inputs (e.g. through feedforward inhibition).

Our derivation of network dynamics directly from our loss function allows us to interpret the properties of this network in terms of neural function. Each spike can be interpreted as a greedy error reduction mechanism—it moves $\hat{\mathbf{x}}$ closer to the signal $\mathbf{x}$. This error reduction is communicated back to the network through recurrent connectivity, thereby reducing the membrane potential of the other neurons. The membrane potential, in turn, can be interpreted as a representation error—it is determined by a linear projection of $\mathbf{x} - \hat{\mathbf{x}}$ onto the neuron's decoding weights, $\mathbf{D}_i$ (*Equation 12*). Following this interpretation, whenever the error becomes too big, the voltage becomes too big and it reaches threshold. This produces a spike, which reduces the error, and so on.

We can also understand these network dynamics in terms of attractor dynamics. This network implements a point attractor—firing rates evolve towards a stable fixed point in N-dimensional firing rate space. The location of this point attractor depends on neural input and network connectivity. When a neuron dies, the point attractor is projected into a subspace given by $r_k = 0$, where neuron $k$ is the neuron that has died.

Note that in this derivation, we used a quadratic cost. This cost increases the value of the spiking threshold (*Equation 13*) and the spiking reset (*Equation 18*). We can also derive network parameters for alternative cost term choices. For example, if we use a linear cost, we simply need to drop the second term ($\beta \delta_{ik}$) in *Equation 18*, while keeping all other parameters the same. In other words, we can implement a quadratic cost by increasing the spiking threshold and the spiking reset, and we can implement a linear cost by increasing the spiking threshold without increasing the spiking reset. In this way, the spiking threshold, and the reset determine the cost function. It is conceivable that these variables may be learned, just as network connectivity may be learned. Alternatively, these values may be predetermined for various brain areas, depending on the computational target of each brain area.

## Derivation of network model with separate excitatory and inhibitory populations

When these ideas are applied to large and heterogeneous networks, we run into one problem concerning their biological interpretation. Individual neurons in these networks can sometimes target postsynaptic neurons with both excitatory and inhibitory synapses, so that they violate Dale's law. To avoid this problem, we consider a network that consists of separate pools of excitatory and inhibitory neurons. The external signals feed only into the excitatory population, and the inhibitory population remains purely local. This scenario is captured by the equations,

$$\frac{dV_i^E}{dt} = -\lambda V_i^E + \sum_{j=1}^{M} F_{ij} g(x_j) + \sum_{k=1}^{N^E} \Omega_{ik}^{EE} s_k^E - \sum_{k=1}^{N^I} \Omega_{ik}^{EI} s_k^I - R_i^E s_i^E + \sigma_V \eta_i, \tag{23}$$

$$\frac{dV_i^I}{dt} = -\lambda V_i^I + \sum_{k=1}^{N^E} \Omega_{ik}^{IE} s_k^E - \sum_{k=1}^{N^I} \Omega_{ik}^{II} s_k^I - R_i^I s_i^I + \sigma_V \eta_i. \tag{24}$$

The four connectivity matrices, $\Omega_{ik}^{EE}$, $\Omega_{ik}^{EI}$, $\Omega_{ik}^{IE}$, $\Omega_{ik}^{II}$ are all assumed to have only positive entries, which enforces Dale's law in this network. The neuron's self-resets are now explicitly captured in the terms $R_i^E s_i^E$ and $R_i^I s_i^I$, and are no longer part of the connectivity matrices themselves. We will now show how to design the four connectivity matrices such that the resulting network becomes functionally equivalent to the networks in the previous section. To do so, we apply the basic ideas outlined in *Boerlin et al. (2013)*, and simplified due to W. Brendel, R. Bourdoukan, P. Vertechi, and the authors (personal communication).

We start with the network in the previous section, as described in *Equation 5*, and split its (non-Dalian) recurrent connectivity matrix, $\Omega_{ik}$, into self-resets, excitatory connections, and inhibitory connections. The self-resets are simply the diagonal entries of $-\Omega_{ik}$, see *Equation 18*,

$$R_i = \mathbf{D}_i^\top \mathbf{D}_i + \beta, \tag{25}$$

where the choice of signs is dictated by the sign conventions in *Equation 23* and *Equation 24*. The excitatory connections are the positive entries of $\Omega_{ik}$, for which we define

$$\Omega_{ik}^{EE} = [-\mathbf{D}_i^\top \mathbf{D}_k - \beta \delta_{ik}]_+, \tag{26}$$

where the notation $[x]_+$ denotes a threshold-linear function, so that $[x]_+ = x$ if $x > 0$ and $[x]_+ = 0$ otherwise. The inhibitory connections are the negative, off-diagonal entries of $\Omega_{ik}$, for which we write

$$W_{ik} = [\mathbf{D}_i^\top \mathbf{D}_k + \beta \delta_{ik}]_+ - \delta_{ik} R_k. \tag{27}$$

With these three definitions, we can re-express the recurrent connectivity as $\Omega_{ik} = \Omega_{ik}^{EE} - W_{ik} - \delta_{ik} R_k$.

Using this split of the non-Dalian connectivity matrix, we can trivially rewrite *Equation 5* as an equation for an excitatory population,

$$\frac{dV_i^E}{dt} = -\lambda V_i^E + \sum_{j=1}^{M} F_{ij} g(x_j) + \sum_{k=1}^{N^E} \Omega_{ik}^{EE} s_k^E - \sum_{k=1}^{N^E} W_{ik} s_k^E - R_i^E s_i^E + \sigma_V \eta_i, \tag{28}$$

which is identical to *Equation 23* above, except for the term with the inhibitory synapses, $W_{ik}$, on the right-hand-side. Here, the inhibitory synapses are multiplied with $s_k^E$ rather than $s_k^I$, and their number is identical to the number of excitatory synapses. Accordingly, this term violates Dale's law, and we need to replace it with a genuine input from the inhibitory population. Since this non-Dalian term consists of a series of delta-functions, we can only replace it approximately, which will suffice for our purposes. We simply require that the genuine inhibitory input approximates the non-Dalian term on the level of the postsynaptic potentials, i.e., the level of filtered spike trains,

$$\sum_{k=1}^{N^E} W_{ik} r_k^E \approx \sum_{k=1}^{N^I} \Omega_{ik}^{EI} r_k^I. \tag{29}$$

Here the left-hand-side is the given non-Dalian input, and the right-hand-side is the genuine inhibitory input, which we are free to manipulate and design.

The solution to the remaining design question is simple. In the last section, we showed how to design arbitrary spiking networks such that they track a given set of signals. Following these recipes, we assume that the inhibitory population output optimally tracks the (filtered) spike trains of the excitatory population. More specifically, we assume that we can reconstruct the instantaneous firing rates of the excitatory neurons, $r_i^E$, by applying a linear readout to the instantaneous firing rates of the inhibitory neurons, $r_j^I$, so that

$$\hat{\mathbf{r}}^E = \sum_{j=1}^{N^I} \mathbf{D}_j^I r_j^I. \tag{30}$$

Note that all entries of $\mathbf{D}_j^I$, the inhibitory decoding weights, must be positive. If the estimate of the excitatory firing rates closely tracks the real firing rates, then we can fulfil *Equation 29*.

To obtain this relation, we will assume that the inhibitory population minimizes its own loss function,

$$E_I = (\mathbf{r}^E - \hat{\mathbf{r}}^E)^2 + \beta^I C(\mathbf{r}^I). \tag{31}$$

where the first term is the error incurred in reconstructing the excitatory firing rates, and the second term is a cost associated with the firing of the inhibitory neurons. Apart from the replacement of the old input signal, $\mathbf{x}$, by the new 'input signal', $\mathbf{r}^E$, this is exactly the same loss function as in *Equation 8*. An important difference is that here the number of input signals, which corresponds to the number of excitatory neurons, may be larger than the number of output spike trains, which corresponds to the number of inhibitory neurons. Our spiking networks will in this case still minimize the mean-square-error above, though the representation could be lossy. Since the effective dimensionality explored by the instantaneous firing rates of the excitatory population is limited by the dimensionality of the input signal space, however, these losses are generally small for our purposes.

Given the above loss function, we can redo all derivations from the previous section, to obtain the connections, thresholds, and resets for the inhibitory population. Defining the short-hand

$$H_{ik}^I = (\mathbf{D}_i^I)^\top \mathbf{D}_k^I + \beta^I \delta_{ik}, \tag{32}$$

we find that

$$\Omega_{ik}^{II} \equiv H_{ik}^I - R_k^I \delta_{ik}, \tag{33}$$

$$\Omega_{ik}^{IE} \equiv [\mathbf{D}_i^I]_k, \tag{34}$$

$$T_k \equiv \frac{H_{kk}^I}{2}, \tag{35}$$

$$R_k^I \equiv H_{kk}^I. \tag{36}$$

Here the first equation is equivalent to *Equation 18*, except for the different sign convention ($\Omega_{ik}^{II}$ enters negatively in *Equation 24*), and for the subtraction of the self-reset term, which is not part of the recurrent connectivity. The second equation is equivalent to *Equation 19*. The third equation is equivalent to *Equation 22*, and the fourth equation is the self-reset term, containing the diagonal elements of *Equation 18*. Note that, due to the positivity of the inhibitory decoding weights, the two connectivity matrices have only positive entries, as presupposed above in *Equation 24*. Consequently, the inhibitory subnetwork obeys Dale's law.

In a last step, we replace the excitatory spike trains in *Equation 28* with their inhibitory approximation. Specifically, we rewrite *Equation 29* as

$$\sum_{k=1}^{N^E} W_{ik} r_k^E \approx \sum_{k=1}^{N^E} W_{ik} \hat{r}_k^E \tag{37}$$

$$= \sum_{j=1}^{N^E} \sum_{k=1}^{N^I} W_{ij} D_{jk}^I r_k^I \tag{38}$$

$$\equiv \sum_{k=1}^{N^I} \Omega_{ik}^{EI} r_k^I, \tag{39}$$

where we used *Equation 30* in the approximation step. This approximation generally works very well for larger networks; in smaller networks, finite size effects come into play.

Finally, when we take into account the results from the previous section, write $\mathbf{D}_i^E$ for the excitatory decoder weight of the $i$-th neuron, and introduce the short-hand

$$H_{ik}^E = (\mathbf{D}_i^E)^\top \mathbf{D}_k^E + \beta^E \delta_{ik}, \tag{40}$$

then we find that the excitatory subnetwork has the connections, thresholds, and resets (compare *Equation 25–Equation 27*),

$$F_{ij} = [\mathbf{D}_i^E]_j, \tag{41}$$

$$\mathbf{\Omega}_{ik}^{EE} = [-H_{ik}^E]_+, \tag{42}$$

$$\mathbf{\Omega}_{ik}^{EI} = \sum_{j=1}^{N_E} \left( [H_{ij}^E]_+ - \delta_{ij}R_j^E \right) \mathbf{D}_{jk}^I, \tag{43}$$

$$T_k^E \equiv \frac{H_{kk}^E}{2}, \tag{44}$$

$$R_k^E = H_{kk}^E. \tag{45}$$

Importantly, the EI network therefore consists of two populations, one excitatory, one inhibitory, both of which minimize a loss function. As a consequence, the excitatory subnetwork will compensate optimally as long as the inhibitory subnetwork remains fully functional. In this limit, the excitatory subnetwork is essentially identical to the networks discussed in the previous section. The inhibitory subnetwork will also compensate optimally until its recovery boundary is reached. In the following, we will focus on one subnetwork (and one loss function again). For notational simplicity, we will leave out the superscript references '*E*'.

## Tuning curves and quadratic programming

For constant input signals, the instantaneous firing rates of the neurons will fluctuate around some mean value. Our goal is to determine this mean value. We will start with the most general case, and rewrite the instantaneous firing rates as $r_i(t) = \bar{r}_i + \eta_i(t, \bar{r}_i)$, were $\eta_i(t, \bar{r}_i)$ is a zero-mean 'noise' term that captures the fluctuations of the instantaneous firing rates around its mean value. Note that these fluctuations may depend on the neuron's mean rate. In turn, neglecting the costs for a moment, we can average the objective function, *Equation 8*, over time to obtain

$$\langle E \rangle_t = \left\langle \left( \mathbf{x} - \sum_i \mathbf{D}_i r_i(t) \right)^2 \right\rangle_t \tag{46}$$

$$= \left( \mathbf{x} - \sum_i \mathbf{D}_i \bar{r}_i \right)^2 + \sum_{ij} \mathbf{D}_i^\top \mathbf{D}_j \langle \eta_i(\bar{r}_i)\eta_j(\bar{r}_j) \rangle_t. \tag{47}$$

For larger networks, we can assume that the spike trains of neurons are only weakly correlated, so that $\langle \eta_i(\bar{r}_i)\eta_j(\bar{r}_j) \rangle_t = \mathrm{Var}(\eta_i(\bar{r}_i))\delta_{ij}$. Here, $\mathrm{Var}(\eta_i(\bar{r}_i))$ is the variance of the noise term. We obtain

$$\langle E \rangle_t = \left( \mathbf{x} - \sum_i \mathbf{D}_i \bar{r}_i \right)^2 + \sum_i \|\mathbf{D}_i\|^2 \mathrm{Var}(\eta_i(\bar{r}_i)). \tag{48}$$

We furthermore notice that the spike train statistics are often Poisson (*Boerlin et al., 2013*), in which case we can make the replacement $\mathrm{Var}(\eta_i(\bar{r}_i)) = \bar{r}_i$. The loss function then becomes a quadratic function of the mean firing rate, which needs to be minimized under a positivity constraint on the mean firing rate. This type of problem is known as 'quadratic programming' in the literature. In this study, we focused on networks for which the contributions of the firing rate fluctuations can be neglected, which is generally the case for sufficiently small readout weights and membrane voltage noise (see *Figure 4—figure supplement 3*). In this case, we obtain the loss function used in *Equation 3*,

$$\langle E \rangle_t = \left( \mathbf{x} - \sum_i \mathbf{D}_i \bar{r}_i \right)^2 + \beta C(\bar{\mathbf{r}}) \tag{49}$$

with the cost term included again.

In general, quadratic programming is mathematically intractable, so the objective function must be minimized numerically. However, in networks with a small number of neurons, we can solve the problem analytically and gain some insight into the nature of quadratic programming. Here, we do this for the two neuron example (*Figure 4A*).

In this example, we assumed that the first neuron contributes with $\mathbf{D}_1 = (w, c)$ to the readout, and the second neuron with $\mathbf{D}_2 = (-w, c)$. Accordingly, the firing rates $\bar{r}_1$ and $\bar{r}_2$ are given by the solution to the following equation:

$$(\bar{r}_1, \bar{r}_2) = \arg \min_{\bar{r}_1 \geq 0, \bar{r}_2 \geq 0} \left[ (x - (w\bar{r}_1 - w\bar{r}_2))^2 + (r_B - (c\bar{r}_1 + c\bar{r}_2))^2 + \beta(\bar{r}_1^2 + \bar{r}_2^2) \right], \tag{50}$$

where $x$ is the variable signal, and $r_B$ is the fixed background signal. The positivity constraint partitions the solution of this equation into three regions, determined by the value of $x$: region $\mathcal{R}_1$ where $\bar{r}_1 = 0$ and $\bar{r}_2 \geq 0$, region $\mathcal{R}_2$ where $\bar{r}_1 \geq 0$ and $\bar{r}_2 \geq 0$, and region $\mathcal{R}_3$ where $\bar{r}_2 = 0$ and $\bar{r}_1 \geq 0$ (*Figure 4—figure supplement 1A*). In region 2, we can then easily solve *Equation 50* by setting the derivative of our loss function to zero. This gives $\bar{\mathbf{r}} = (\mathbf{w}^\top \mathbf{w} + \mathbf{c}^\top \mathbf{c} + \beta \mathbf{1})^{-1} (\mathbf{w}x + \mathbf{c}r_B)$ where $\mathbf{w} = (w, -w)$, $\mathbf{c} = (c, c)$, and $\mathbf{1}$ is the identity matrix. Looking at this equation, we see that $\bar{r}_1 = 0$ when $x = -cr_B/w$ and $\bar{r}_2 = 0$ when $x = cr_B/w$. Therefore, the firing rate solution for region two is valid when $-cr_B/w \leq x \leq cr_B/w$. For $x \geq cr_B/w$, we have $\bar{r}_2 = 0$ because of the positivity constraint in *Equation 50*. This is region 3. We can calculate $\bar{r}_1$ by setting $\bar{r}_2 = 0$ and then minimizing the loss function. This gives $\bar{r}_1 = (w^2 + c^2 + \beta)^{-1}(wx + cr_B)$. Similarly, for $x \leq -cr_b/w$ we have $\bar{r}_1 = 0$ because of the positivity constraint. This is region one and we obtain $\bar{r}_2 = (w^2 + c^2 + \beta)^{-1}(-wx + cr_B)$. The firing rates within each region are given by a simple linear projection of $\mathbf{x}' = (x, r_B)$, although the size and direction of this projection is different in each region. As such, the solution to this quadratic programming problem is a piece-wise linear function of $x$.

In networks with larger numbers of neurons, the solution will still be a piece-wise linear function of $x$, although there will be more regions and the firing rate solutions are more complicated because more neurons are simultaneously active. In contrast, the transformation from firing rates to $\mathbf{x}'$ is very simple (*Equation 6*). It is given by a simple linear transformation, and is region independent (*Figure 4—figure supplement 1B*).

## Readout weights and cost terms: 1-d and 2-d example

There are a number of free parameters in our model, such as the cost terms and the readout weights. The choice of these values determine the precise shape of tuning curves. In general, however, the precise values of these terms have little influence on the coding capability of our system, once certain properties have been satisfied, which we will outline here.

The cost term that we used for the examples in *Figures 1–6* is a quadratic cost, $C(\mathbf{r}) = \sum_k r_k^2$. This cost term encourages the system to find a solution in which all neurons share in the signal representation. (Here, and in the following, we will no longer distinguish $r_i$, the instantaneous firing rate used in the spiking network, and $\bar{r}_i$, the mean firing rates used in quadratic programming. The appropriate notation will be determined by the context.) The cost term for *Figures 7–9*, the V1 model, is a linear cost, $C(\mathbf{r}) = \sum_k r_k$. This cost term limits the overall number of spikes, and generally encourages sparse solutions, in which most neurons contribute very little to the signal representation. Regardless of this choice, our general predictions about optimal compensation are qualitatively similar, as long as the cost term does not dominate the loss function.

The other important parameters that we must choose are the decoding or readout weights. In *Figure 2*, the decoding weights of the excitatory population were drawn from a Gaussian distribution with mean $\mu_E = 2$, and standard deviation $\sigma_E = 0.2$. For the inhibitory population, we used a mean $\mu_I = 0.3$ and a standard deviation $\sigma_I = 0.03$. For the networks in *Figures 3–6*, we have used regularly spaced decoding weights, with the addition of some random noise. The parameter values that we used are plotted in *Figure 3D*, *Figure 4A–C*, left column, and *Figure 5A,E,I*.

All of our quadratic programming and optimal compensation predictions for tuning curve shapes still hold for alternative choices of readout weights, once the following properties are satisfied. First, it is important that the readout vectors span the space of the signal that we want our network to represent. Otherwise, the system will not be able to represent signals along certain directions, or compensate for neuron loss. Second, it is important to set the scale of the readout, so that the cost does not dominate. There is a natural scaling that we can use to avoid such problems. We require the size of firing rates and the readout to be independent of network size. From $\hat{x}_j = \sum_k^N D_{jk} r_k \sim \mathcal{O}(1)$, it follows that $D_{jk} \sim \mathcal{O}(1/N)$. As a consequence, the off-diagonal elements of the recurrent connectivity are small $\Omega_{ik} \sim \mathcal{O}(1/N^2)$, for $i \neq k$ (*Equation 18*), and if we assume that the diagonal elements are on the same order of magnitude, then $\beta \sim \mathcal{O}(1/N^2)$. This scaling provides a principled basis for parameter choices in our model.

We may also want to scale our decoder weights without changing the shape of our tuning curve prediction (*Figure 4—figure supplement 3B*). To do this, the cost parameter $\beta$ and the membrane potential leak $\lambda$ must also be scaled together. Specifically, if the readout weights are given by $\{\alpha \times D_{jk}\}$, where $\alpha$ is a scaling parameter that characterizes the size of the decoder weights and $\{D_{jk}\}$ are fixed decoder weights, then the spiking cost parameter must be set to $\alpha^2 \times \beta$ and the membrane potential leak must be set to $\alpha \times \lambda$. We can see that this preserves the shape of tuning curves by looking at the resulting structure of our loss function (*Equation 2*):

$$E(\mathbf{r}) \;\; = (\mathbf{x} - (\alpha \mathbf{D}) \cdot \mathbf{r})^2 + \alpha^2 \beta \sum_{i=1}^{N} r_i^2 \,. \tag{51}$$

As before, the minimum of this loss function gives firing rates in units of the inverse membrane potential leak $(\alpha \lambda)$. Therefore, we must divide $\mathbf{r}$ by $(\alpha \lambda)$ to obtain firing rates in units of Hz. Our loss function then becomes:

$$E(\mathbf{r}/(\alpha \lambda)) = (\mathbf{x} - (\alpha \mathbf{D}) \cdot \mathbf{r}/(\alpha \lambda))^2 + \alpha^2 \beta \sum_i (r_i/(\alpha \lambda))^2 \tag{52}$$

$$= (\mathbf{x} - \mathbf{D} \cdot \mathbf{r}/\lambda)^2 + \beta \sum_i (r_i/\lambda)^2 \,. \tag{53}$$

This loss function is independent of $\alpha$, and so, using this scaling the optimal tuning curves will have the same shape for all values of $\alpha$.

## Readout weights and cost terms: V1 example

There are many possible choices of decoder weights $\{\mathbf{D}_i\}$ that provide a faithful representation of a signal. In positive sparse coding, we choose the decoder weights that provide the most efficient signal representation, for a sparse cost term ($C(\mathbf{r}) = \sum_k r_k$), under the constraint that firing rates must be positive. Here, we describe how we calculate these positive sparse coding weights, which we will use in several of our figures (*Figures 7–9* and *Figure 9—figure supplement 1–2*).

We use the signal vector $\mathbf{x} = (x_1, \dots, x_j, \dots, x_M)$ to denote an image patch, where each element $x_j$ represents a pixel from the image patch (*Olshausen and Field, 1996*). We quantify the efficiency of a sparse representation using the following loss function:

$$E = \left\langle (\mathbf{x} - \hat{\mathbf{x}})^2 \right\rangle + \beta \left\langle \sum_i r_i \right\rangle, \tag{54}$$

where $\langle ... \rangle$ denotes an average across image patches. This is simply *Equation 2* with a sparse cost term, averaged over image patches. The first term in this loss function is the image representation error. The second term quantifies the sparsity of the representation. The decoding filters that minimize this loss function will be the positive sparse coding filters for natural images.

We assume that the decoding filters are optimized to represent natural images, such as forest scenes, flowers, sky, water and other images from nature. Natural images are chosen because they are representative of the images that surround animals throughout evolution. We randomly select 2000 image patches of size $12 \times 12$ from eight natural images taken from Van Hateren's Natural Image Database (*van Hateren and van der Schaaf, 1998*). These images are preprocessed by removing low-order statistics, so that our sparse coding algorithm can more easily learn the higher-order statistical structure of natural images. First of all, images are centered so as to remove first order statistics:

$$\mathbf{x} \rightarrow \mathbf{x} - \langle \mathbf{x} \rangle. \tag{55}$$

Next, images are whitened, so as to remove second-order statistics:

$$\mathbf{x} \rightarrow \mathbf{M}^{-1/2} \mathbf{x}, \tag{56}$$

where $\mathbf{M} = \langle \mathbf{x}\mathbf{x}^T \rangle$ is a decorrelating matrix.

We calculate sparse coding filters by minimizing the loss function (*Equation 54*) using a two step procedure. First, for each image patch in a subset of 50 image patches, we calculate the firing rates that minimize the loss function under the constraint that firing rates must be positive:

$$\mathbf{r} = \arg \min_{r_i \geq 0 \, \forall i} E.$$ (57)

The positivity constraint reduces the representational power of the neural population, so, to counteract this, we use a large population containing twice as many neurons as signal dimensions, $N = 2M$ and we initialize our decoder matrix to $D_{jk} = \delta_{j,k} - \delta_{j,k+M}$ to ensure that our neural population can easily span the signal space throughout the sparse coding calculation.

Next, we update our decoding vectors by stochastic gradient descent on the loss function, using the optimal firing rates calculated in the previous step:

$$\mathbf{D}_i \rightarrow \mathbf{D}_i - \epsilon \frac{dE}{d\mathbf{D}_i} = \mathbf{D}_i + \epsilon 2 \langle (\mathbf{x} - \hat{\mathbf{x}}) r_i \rangle.$$ (58)

Decoding weights are normalized for each neuron, so that they do not become arbitrarily large. This calculation is similar to sparse coding calculations performed before (*Olshausen and Field, 1996*), except that we enforce the additional requirement that firing rates must be positive. This constraint was used before for sparse non-negative matrix factorization (*Hoyer, 2003*, *2004*). However, this method requires an additional constraint that decoding weights be strictly positive, so that the non-negative matrix factorization method can be applied. This additional constraint will reduce the efficiency of the representation, because it precludes solutions that may be more efficient, but violate the additional constraint.

To compare the properties of optimal compensation in this positive sparse coding model to experiment, we calculate the direction tuning of neurons in our model using a protocol similar to that used in the experimental work (*Crook and Eysel, 1992*). Specifically, we drive our model using a directed, edge-like stimulus. These stimuli are Gabor filters, with a profile similar to the sparse coding filters. We calculate the firing rate response of each neuron using *Equation 57* for 16 different stimulus directions. For each direction, the Gabor filter is positioned at regularly spaced locations along a line perpendicular to the Gabor filter orientation and the maximum firing rate of each neuron along this line is taken as the firing rate response of that neuron at that direction. These firing rate responses form the direction tuning curves that we can compare to experimental recordings.

## The impact of neuron loss

We investigate the impact of neuron loss in our spiking model by simulating the network both before and after neuron loss (*Figure 2* and *Figure 3*). Specifically, we use the Euler method to iterate the dynamics described by *Equation 17* or *Equation 23*, *Equation 24*, and we measure the readout $\hat{x}$, the spike trains $\mathbf{s}$, and the membrane potentials $\mathbf{V}$. We simulate the loss of a neuron by setting all the connections to and from that neuron to zero. This is equivalent to silencing a neuron. We continue to measure the network activity and the readout after neuron loss.

We also calculate the impact of neuron loss on tuning curves with and without optimal compensation (*Figure 5*). To calculate tuning curves without compensation we solve *Equation 3* in the intact state. We then constrain the firing rates of neurons selected for loss to zero and calculate the impact of this on our signal representation. To calculate the impact of neuron loss with optimal compensation, we solve *Equation 4*.

The integrate-and-fire model that we have described can produce spikes at arbitrarily large firing rates. This can be a problem, especially when neurons die and the remaining neurons compensate with extremely high, unrealistic firing rates. To avoid this, we can include a form of adaptation in our spiking model. Specifically, we can extend our spiking rule so that a neuron $i$ will only spike if its firing rate $f_i$ is lower than a maximum $f_{max}$. Here, the firing rate $\mathbf{f}$ is given by the following differential equation:

$$\frac{d\mathbf{f}}{dt} = -\mathbf{f}/\tau_A + \mathbf{s},$$ (59)

and $\tau_A$ is the time scale of adaptation. This time scale is much slower than the time scale of

the spiking dynamics τ. We use this extended neural model to calculate the recovery boundary in *Figure 3J,K*, where we kill neurons in random order.

In turn, we can compute the firing rates and recovery boundary of this extended integrate-and-fire model using quadratic programming (data not shown). Specifically, the firing rates of this network are given by the solution of the following optimization problem:

$$\mathbf{r}(\mathbf{x}) = \arg\min_{R} E(\mathbf{x}), \tag{60}$$

where $R$ denotes the set of constraints that the firing rates must satisfy:

$$R = \left\{ r_{max} \geq r_i \geq 0 \, \forall i; r_j = 0 \, \forall j \in X \right\} \tag{61}$$

and where $X$ is the set of all dead (or silenced) neurons.

To study the effects of redundancy (or overcompleteness) in the V1 model (*Figure 9—figure supplement 2A*), we define an over-completeness factor, $K$, given by $K = N/M$, where $N$ is the number of neurons in the representation and $M$ is the signal dimension. In the sparse coding calculation above, we had $K = 2$. To increase the redundancy or overcompleteness of the representation, we generated a new set of readout vectors by two methods. In method one, we randomly drew $N = KM$ decoding vectors $\mathbf{D}_i^{(K)}$ from the V1 model and then added multiplicative noise, in order to avoid parallel readout weights. Specifically, the new readout vectors $\mathbf{D}_i^{(K)}$ were obtained from the old readout vectors, $\mathbf{D}_i^{(2)}$, using $\mathbf{D}_{i+a}^{(K)} = \mathbf{D}_i^{(2)} \cdot (1 + \boldsymbol{\xi}_{i+a})/(K/2)$, where $i \in [1, \ldots, 2M]$, $\xi_{ij} \sim \mathcal{N}(0, 0.1)$, $a = 2Mm$ and $m \in [0, \ldots, K/2 - 1]$. In method two, we also randomly drew decoding vectors from the V1 model, but then shifted the corresponding $12 \times 12$ image patch randomly in space. Both methods yielded similar results. We used these methods to calculate readout weights for over-complete representations because the sparse coding calculation becomes computationally prohibitive for large values of $M$.

## Tight balance of excitation and inhibition

In simulations of our spiking model, we see that optimal coding generally coincides with a balance of excitatory and inhibitory currents into each neuron (*Figure 2G* and *Figure 3H,I*). To characterize this balance, we define three currents for each excitatory neuron. First, we define the excitation $E_i$ as the total input current received through excitatory synapses. Second, we define the inhibition $I_i$ as the total input current received through inhibitory synapses. Third, we define the reset current $\bar{R}_i$ as the total current caused by a neuron's self-reset after an action potential. Since our idealized neuron model uses delta-functions to capture the synaptic currents evoked by single spikes, we smooth the delta-functions in order to obtain more realistic synaptic currents. For simplicity, we use the same filter as in the computation of the instantaneous firing rates, although other filters could be used as well. In the EI network, *Equation 23* and *Equation 24*, the three currents into the $i$-th excitatory neuron are then given by

$$E_i = \sum_{k=1}^{N} \Omega_{ik}^{EE} r_k^E + \sum_{j=1}^{M} F_{ij}^E x_j, \tag{62}$$

$$I_i = \sum_{k=1}^{N} \Omega_{ik}^{EI} r_k^I + \sum_{j=1}^{M} F_{ij}^I x_j, \tag{63}$$

$$\bar{R}_i = R_i^E r_i^E, \tag{64}$$

where $F_{ij}^E \equiv F_{ij}$ if $F_{ij} x_j > 0$ (and $F_{ij}^E \equiv 0$ otherwise), and $F_{ij}^I \equiv F_{ij}$ if $F_{ij} x_j < 0$ (and $F_{ij}^I \equiv 0$ otherwise). When the inhibitory subpopulation is not explicitly simulated, we simply set $\Omega_{ik}^{EE} r_k^E \equiv \Omega_{ik} r_k$ unless $\Omega_{ik} < 0$ and $\Omega_{ik}^{EI} r_k^I \equiv \Omega_{ik} r_k$ unless $\Omega_{ik} > 0$.

Using the above definitions of currents, we can integrate *Equation 23* to re-express the voltage of an excitatory neuron as $V_i^E = E_i - I_i - \bar{R}_i$. This voltage is trivially bounded from above by a neuron's threshold, $V_i^E < T_i$. Due to the design properties of the network, the voltage is also bounded from below: since the voltage reflects a representation error, neurons with decoding vectors opposite to those of neuron $i$ will keep negative errors in check. If we neglect cost terms, we obtain

$V_i^E > R_i^E$. (Otherwise the bound is lower). Re-expressed for the EI ratio, we obtain the following bounds

$$1 + \frac{\bar{R}_i - R_i^E}{I_i} < \frac{E_i}{I_i} < 1 + \frac{\bar{R}_i + T_i}{I_i}. \tag{65}$$

The EI ratio will therefore approach one (perfect balance) if the inhibitory synaptic currents are much larger than the self-reset currents and voltage thresholds, or, if excitatory currents are cancelled by inhibitory currents rather than by a neuron's spikes and subsequent self-resets.

The network does not reach this balanced regime under two conditions. First, if a network has a small redundancy, i.e., the number of neurons, $N$, is not much larger than the number of input signals, $M$, then each neuron carries a large load of the representation, and the reset currents become similar in magnitude to the synaptic currents. This is the case for the cricket cercal system, for instance, for which $N = 4$ and $M = 2$. Second, if a system approaches the recovery boundary, the representational burden of some neurons also increases a lot, leading to a similar effect. This is the case for the green neuron in *Figure 3I*, even after the first k.o, and leads to a more regular firing pattern. In other words, neurons that compensate for neuron loss can sometimes experience excessive excitatory currents, even if the compensation is successful.

To capture the mechanistic difference between compensation and recovery boundary more clearly, we therefore also defined the ratio of positive to negative currents, $E_i/(I_i + \bar{R}_i)$ (*Figure 3H*). This ratio is bounded by

$$1 - \frac{R_i^E}{I_i + \bar{R}_i} < \frac{E_i}{I_i + \bar{R}_i} < 1 + \frac{T_i}{I_i + \bar{R}_i}. \tag{66}$$

and remains close to one as long as there are sufficiently large negative currents, or, if excitatory currents are cancelled by either recurrent inhibition or by spiking. As long as the network is fully functional, the latter will always be the case. Once the recovery boundary is hit, however, the lower bound on the voltages of some neurons disappears, and so will the lower bound on the above ratio of positive to negative currents (as well as the EI ratio), which is the effect we observe in *Figure 3HI*.

Traditionally, balance is often understood in the limit of very large networks (*van Vreeswijk and Sompolinsky, 1996*, *1998*), and these classical scaling arguments can also be applied to our networks (*Boerlin et al., 2013*). Neglecting Dale's law once more for simplicity, we can increase the network size, while keeping the relative size and number of the input signals and the neuron's firing rates constant. Since $\hat{x}_j = \sum_k D_{jk} r_k \sim \mathcal{O}(1)$ and $r_k \sim \mathcal{O}(1)$, we find that $D_{jk} \sim \mathcal{O}(1/N)$ and $\beta \sim \mathcal{O}(1/N^2)$. As a consequence of *Equation 18* and *Equation 13*, we observe that both the recurrent connection strengths and the spiking thresholds are small, so that $\Omega_{ik} \sim \mathcal{O}(1/N^2)$ and $T_i \sim \mathcal{O}(1/N^2)$. Now, because recurrent input is summed across the entire population of $N$ neurons, we find that $V_i \sim \mathcal{O}(N/N^2) \sim \mathcal{O}(1/N)$, and similar observations hold for $I_i$, $E_i$, and $\bar{R}_i$. Hence, these currents are $N$ times larger than the spiking threshold, $T_i$, or the reset, $R_i$. The EI ratio in *Equation 65* then approaches one, and excitatory and inhibitory currents cancel precisely. We note that this balance of excitation and inhibition is much tighter than in balanced networks with random connectivity (*van Vreeswijk and Sompolinsky, 1996*, *1998*). In randomly connected networks the balance between excitation and inhibition is of order $\mathcal{O}(1/\sqrt{N})$ (*van Vreeswijk and Sompolinsky, 1998*), whereas here these fluctuations are $\mathcal{O}(1/N)$. We note that these observations only hold when the number of input signals, $M$, is kept constant; if that number grows with $N$, then the scaling is different. The conceptual differences between loose and tight balance were recently highlighted in a review (*Denève and Machens, 2016*).

## Optimal compensation proof

The spiking network that we use is capable of rapidly implementing optimal compensation, without requiring any synaptic plasticity mechanisms. We can prove this by showing that an optimal network of $N - 1$ neurons is equivalent to an optimal network of $N$ neurons after the loss of one neuron. For the EI network, the argument applies separately to the excitatory and inhibitory subnetworks.

For the sake of argument, we suppose that the $N^{th}$ neuron dies. At a mechanistic level, the loss of this neuron is equivalent to cutting all the connections to and from the dead neuron and from the

dead neuron to the readout $\hat{\mathbf{x}}$. Therefore, a network where the $N^{th}$ neuron has died is equivalent to a network with $N-1$ neurons with readout matrix $D_{ij}^* = D_{ij}\,\forall j \in [1, N-1]$ and $\forall i \in [1, M]$, with feedforward connectivity $F_{ij}^* = F_{ij}\,\forall i \in [1, N-1]$ and $\forall j \in [1, M]$, with recurrent connectivity $\Omega_{ik}^* = \Omega_{ik}\,\forall i, k \in [1, N-1]$ and with spiking thresholds $T_i^* = -\Omega_{ii}/2\,\forall i \in [1, N-1]$.

Now, we compare this damaged network to an optimal network consisting of $N-1$ neurons. To make a fair comparison, we assume that this network has the same readout matrix $\mathbf{D}' \equiv \mathbf{D}^*$ as the reduced damaged network. Then, the recurrent connectivity for this network is given by $\mathbf{\Omega}' = \mathbf{\Omega}^*$, the feedforward connectivity is given by $\mathbf{F}' \equiv \mathbf{F}^*$ and the spiking thresholds are given by $\mathbf{T}' \equiv \mathbf{T}^*$. This configuration is equivalent to the reduced damaged network. Therefore, a spiking neural network whose neurons are individually tuned to represent a signal optimally before cell loss will perform optimal compensation and provide an optimal signal representation after cell loss.

## Acknowledgements

We thank Tony Movshon, Pedro Gonçalves, and Wieland Brendel for stimulating discussions and Alfonso Renart, Claudia Feierstein, Joe Paton, and Michael Orger for helpful comments on the manuscript. S.D. acknowledges the James McDonnell Foundation Award and EU grants BACS FP6-IST-027140, BIND MECT-CT-20095–024831, and ERC FP7-PREDSPIKE. C.K.M. acknowledges an Emmy-Noether grant of the Deutsche Forschungsgemeinschaft and a Chaire d'excellence of the Agence National de la Recherche.

## Additional information

### Funding

| Funder | Grant reference number | Author |
| --- | --- | --- |
| Deutsche Forschungsgemeinschaft | Emmy-Noether | Christian K Machens |
| Agence Nationale de Recherche | Chaire d'Excellence | Christian K Machens |
| James S. McDonnell Foundation | | Sophie Denève |
| European Research Council | ERC FP7-PREDSPIKE | Sophie Denève |
| European Research Council | BIND MECT-CT-20095-024831 | Sophie Denève |
| European Research Council | BACS 796 FP6-IST-027140 | Sophie Denève |

The funders had no role in study design, data collection and interpretation, or the decision to submit the work for publication.

### Author contributions

DGTB, CKM, Designed the study, Performed the analysis, Wrote the manuscript; SD, Designed the study

### Author ORCIDs

Christian K Machens, http://orcid.org/0000-0003-1717-1562

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
