## [Decision Letter]

Thank you for submitting your work entitled "Optimal compensation for neuron death" for consideration by *eLife*. Your article has been favorably evaluated by Eve Marder (Senior Editor) and three reviewers, one of whom is a member of our Board of Reviewing Editors.

The reviewers have discussed the reviews with one another and the Reviewing Editor has drafted this decision to help you prepare a revised submission.

All of the reviewers felt that the submitted theoretical work on compensating for cell death was novel and interesting. However, it was also felt that many specifics were unclear or buried in other places, making biological interpretation difficult (e.g., excitatory, inhibitory cell types). The reviewers encourage the authors to consider the five following points, and at least the first four should be addressed in a revised submission. Additional points raised by the reviewers are detailed below, and should also be addressed.

1) State clearly that the network is non-Dalian and explain the reasoning for this choice.

2) State clearly what the recovery boundaries are, and plot figures that show which variables affect the recovery boundary for a given network with a given task. In general, spend a bit more time on the general mechanism before going into the additional models, or alternatively, explain in detail the additional models, like e.g. Figure 7, which is devoid of explanatory power.

3) An example of a Dalian network, independently of whether it works (same boundary?) or not (why does it not work?) should be added.

4) Show the compensatory mechanisms at play more carefully in figures, i.e. show how balance is changing and finally breaking at the recovery boundary.

5) Consider building a reservoir network model as in Sussillo & Abbott trained to perform a task. Delete neurons, and check whether it is able to self-balance. If it doesn't, address whether this is related with the complexity of the task.

We are including the original reviews as they provide context for the revision points shown above.

*Reviewer #1:*

In this clearly written model theory paper, the authors propose that neural circuits compensate for neuronal death immediately, and show this using firing rate network models and several experiments. Although overall I found the work compelling, in general, I felt that the authors need to better describe 'biological correlates' for their models and to 'fill in the blanks' in various places to help the reader relate and better appreciate the work.

Specifically:

1) 'quadratic programming algorithm' and 'loss function' – could the authors provide suggestions of biological correlates of this. Presumably it would relate to excitatory-inhibitory balances in some way?

2) While spiking rate models are used, differences between excitatory and inhibitory cells are not apparent to me. Given this, then the suggestion in the Discussion that 'these predictions could be tested using neuronal ablations or optogenetic silencing' would seem to suggest that cell type (at least as far as excitatory or inhibitory) does not matter, but this is clearly not true (one of many examples would be epilepsy work by Soltesz group – Krook-Magnuson et al. 2013 Nature Communications). For the reader to better appreciate what is intended by the various discussion suggestions, the authors should 'fill in the blanks' between their theoretical model and experimental suggestions being made.

In the Methods it is stated that "To be biologically more plausible, however, the derivative could be computed through a simple circuit that combines direct excitatory signal inputs with delayed inhibitory signal inputs (e.g. through feedforward inhibition)."

So, could the authors provide description/interpretation of excitatory/inhibitory cells/circuits in terms of their results/predictions? (i.e., biological plausibility).

3) The authors use only positive firing rates (unlike other theoretical studies with positive and negative etc.), and this is a critical difference in what they achieve. While it is clear that negative firing rates may not make sense, the biological correlate of the authors' model is not clear – see 2.

*Reviewer #2:*

Barret et al. investigate a question rarely addressed in theoretical neuroscience: how do neural circuits compensate the death of neurons? They provide an elegant model showing that neurons in the balanced networks can dynamically adjust their rates so as to minimize the impact neuron death. They show how to theoretically compute the optimal tuning curves of neurons in a network encoding a multi-dimensional variable x as a quadratic minimization problem with only constraint that firing rates must be positive. They continue to show that the method presented in their previous work (Boerling et al. 2013) in which they build an integrate-and-fire network optimized to minimize a cost function quantifying the estimation error of an input signal can in fact implement these optimal tuning curves almost exactly. They investigate how the tuning curves change when a fraction of the neurons are silenced and validate some of the predictions derived from the model with three data sets: the goldfish oculomotor system, the cercal system of crickets/cockroaches, V1 orientation tuned neurons. The paper nicely connects two distinct approaches in theoretical neuroscience, network dynamics and efficient coding, and obtains some very general results showing how constraints such as neuronal silencing or more simply, the rectification of the rate, affect the efficient encoding of external variables in a network of spiking neurons. The methodology and the findings presented can potentially have a substantial impact in our understanding of the properties of balanced networks and population encoding. There are however some aspects that the authors need to address in order to present the limitations of the model and the comparison with the data more clearly (see Comments below).

Comments:

Dale's law and the explanation of the EI balance breakdown. In Figure 2, the authors illustrate how when a fraction of the neurons below the recovery boundary is inactivated, the network dynamically compensates and keeps encoding x(t): the remaining neurons change their firing rate and the EI balance is maintained. Then they show that when all neurons with positive readout weight are removed, the balance can no longer be maintained because "there are no longer enough inhibitory cells to balance excitation (or enough excitatory cells to balance inhibition)". This is a very misleading statement given that in this network there are no excitatory or inhibitory neurons (i.e. does not respect Dale's principle). I was very confused when I read this example thinking that neurons in the Figure should have been organized according to whether they are E or I, and not according to the sign of their readout weight. I had to go to the Methods section to later find out that there are no E and I cells in this network. This is in my opinion a main drawback given that (1) Dale's law seems to be true in cortical circuits and (2) the authors are presenting their work as an extension of classical work describing EI networks in which E and I populations are distinct. In their previous manuscript (Boerling et al. 2013) they address this question and propose potential solutions. All of it should be explicitly mentioned and discussed. Moreover, when describing Figure 2 it should be explained why cells with positive readout weights (w+) can naively be viewed as excitatory cells when they provide both excitation and inhibition? Without this most readers will be misled to interpret that w+ neurons are Excitatory and w- neurons are inhibitory.

Biased cost. The authors use a biased quadratic cost function containing a second term which tries to maintain a particular read-out of the rates (coefs c_k_) equal to a constant that they call the background rate. The authors describe in much detailed the implication of this bias in the intercept of the tuning curves (Figure 3 and Figure 3—figure supplement 2B-C). The motivation for this choice is however very succinct. Is there any benefit on having a non-zero intercept regarding the estimation error? I ask this because it is a fairly common observation that cortical circuits seem to maintain the overall average population firing rate constant when dynamically encoding different variables (see e.g. Figure 2 Fujisawa et al., Nat Neuroscience 11 (7), 2008). This observation seems to be obtained to the network in which there is a non-zero r_B_ and non-zero uniform c_k_'s which seems interesting. However, the implications of this additional bias regarding the encoding ability of the circuit (estimation error of x(t)), its plausibility regarding the available data, etc. and not discussed.

The prediction on the changes in tuning for V1 neurons is only shown in three units (Figure 8). In Figure 8—figure supplement 1, the full population of neurons is examined but the changes are only reported in response to the "preferred stimulus orientations". Thus, evaluating the prediction that there should be an increase of firing rate on the non-silenced neurons in response to the "preferred orientation of the silenced cells" is not possible. Either a different population analysis is performed or the statement about the outcome of the prediction at the population level (Results, last paragraph) should be modified.

Results on the comparison with data should be brought forward. Since the paper is quite long as it is now, and the most novel and most interesting part is the comparison of the model predictions with data from three systems (Figure 5–Figure 8), I suggest moving the mechanistic part of how to implement the optimal solution using a spiking network (e.g. Figure 1–Figure 2) to an Appendix box or some sort of "side note figure". This mechanistic part is an important aspect but, once it has been shown that it works, which has somehow already been covered in Boerling et al. 2013, it is no longer required to make the comparison with data (Figure 5–Figure 8, which are all derived by numerically solving Eqs. 4-5). Doing so, the reader will be able to quickly reach the most novel aspects without the need to go over the network implementation details.

In the Discussion, the authors state that the strongest prediction of the model is that "neurons with similar tuning to the dead neurons increase their firing rates and neurons with dissimilar tuning decrease their firing rates (unless they are already silent)." The second part of this statement seems at odds to what is shown in Figure 6, Figure 8 (top) and Figure 8 where dissimilar neurons either don't change much (e.g. Df is always >=0 in Figure 8) or tend to increase their rate in response to the preferred orientation of the dead cells (Figure 8 top). This rate decrease in dissimilar neurons seems to occur in the network shown in Figure 4 with monotonically increasing tuning curves. The details about the changes predicted and the generality of these changes in different systems (bell-shaped vs. monotonic tuning curves) need to be clarified.

At the end of the Methods, the authors provide an explanation of the Tight balance between excitation and inhibition and a scaling argument that concludes that in the network consider here, the cancellation between the mean excitatory and inhibitory drives occurs with precision 1/N. They compare this cancellation with the classical balanced network presented in van Vreeswijk and Sompolinsky (1998) where the cancellation occurs up to order 1/sqrt(N). This comparison is interesting but in order to make it the authors should provide some more information about the behavior of the network in the large N limit: if x is constant and independent of N, do the firing rates converge to a constant value as N increases? This seems hard to visualize as, even when counting on the cancellation of the mean E and I inputs, the magnitude of the total input current fluctuations σ will grow with respect to the threshold (σ~1/N^2/3 whereas threshold is 1/N^2). This is a reflection of the fact that the rate of incoming spikes grows with N whereas the size of each PSP is always of the order of the threshold (i.e. 1/N^2). I don't see how this network can asymptotically converge to non-zero rates in this large N limit.

A final comment: the authors present as an advantage the speed of the compensation mechanism which they say is "faster than the timescale of neural spiking". To me this comment remarks the mismatch of time-scales between the two processes: (1) neural death, an infrequent event, relative to the time scale of neurons, and (2) balanced dynamics, almost instantaneous. More than a bonus I see it as an indication that there are probably other slower processes at work when compensating for such an irreversible damage that happens in such long time-scales. Presented as a prediction for manipulation experiments, the compensation can be viewed as the signature mechanism of balanced dynamics, as done here in the comparison with the experimental data. Additionally, there might be circumstances in which groups of neurons become transiently and frequently inactive (e.g. local DOWN state). The fast compensation described here would seem more tuned to address those fast inactivation events instead of neural death.

*Reviewer #3:*

The manuscript at hand describes a phenomenon that is a routine occurrence for any living nervous system, namely the loss of some of its neurons through cell death. Even in the most severe cases of diseases like presumably Alzheimer's, such neuronal losses often go unnoticed for prolonged periods of time, presumably because they are A) tolerable in a system that relies on massively redundant population code, or B) the system can quickly compensate for the loss of neurons as they fall silent. Due to the lack of cortical plausible, mechanistically implementable computations, the modelling community has not approached this interesting research topic with much vigour.

Here, Barret et al. pick up from the well-described recurrently balanced neural network literature and study how balanced networks of various sizes compensate for neuron loss. They set out with two simple assumptions, i.e. that (1) outside stimuli can be described by combinations of quasi-steady-state firing rates of all neurons in the network and that (2) these firing rates are bounded by a cost function that prevents combinatorial codes in which the majority of neurons are silent in favour of a few active ones. With these two simple ingredients the authors show that the total number of spikes remains constant in the face of dramatic cell losses, as long as a tight balance of excitation and inhibition in the remaining neurons can be maintained. In fact, it is this balance that, when disturbed by cell loss, increases or decreases the firing rates in the remaining neurons to maintain faithful cumulative signal representation. The authors go on to show that this simple feature of balanced networks can account for compensatory behaviour in more complicated models such as Ben Yishai and Sompolinsky's (1995) bump attractor model (that is unfortunately not cited) and even more laborious models of sparse coding in V1. The results evoke vacillating responses in this reviewer of alleging outrageous triviality (anyone modelling balanced systems intuitively knows that this compensation happens) and respecting the originality of the thought process behind the sequence of presented results. After some deliberation, I agree with the authors fully in that optimal compensation by means of shifting, detailed balance has not been proposed before and is a worthwhile idea to entertain.

To show the link of shifting balanced dynamics in response to cell death, I would like to ask the authors to include better explanations and figures panels for all models that exhaustively show the mechanistic origins of the phenomenon they showcase, and not just its effects, i.e. the final tuning curves before and after. Additionally, I would wish for a better graphical way to discern between the two cases of cell loss on either side of the recovery boundary. In my opinion, Figure 3, Figure 4, Figure 5 could be integrated into Figure 2 or outsourced to supplementary materials and give way for the interesting cases of Figure 6–Figure 8 without much loss of detail. Lastly, I would like to ask the authors to discuss clearly what experimental results could discriminate the initially mentioned scenarios A and B (redundancy and compensation, respectively), and what they would expect in the case of temporally varying neuronal representations such as the en vogue neuronal dynamics of Churchland and Shennoy (2012).

[Editors' note: further revisions were requested prior to acceptance, as described below.]

Thank you for resubmitting your work entitled "Optimal compensation for neuron loss" for further consideration at *eLife*. Your revised article has been favorably evaluated by Eve Marder (Senior Editor) and two reviewers, one of whom is a member of our Board of Reviewing Editors.

The authors have addressed four of the five suggestions we made in the first round of reviews, and the paper has improved. However, the fundamental logic of the paper is still not quite clear and needs to be made transparent. Specifically, the Methods section is hard to follow, and there seem to be some discrepancies in updating (e.g., it refers to a Figure 2 that does not exist, and Figure 2 is not referred to in the text). The derivation and translation into Dalian spiking networks is far from clear. Further, please note the following four points:

1) They create what they call "tightly balanced" networks, i.e. networks with very fast, precise / detailed inhibition by means of mathematical derivation and go on to say that the networks they built are much more balanced than vanVreesvijk models. We cannot assess this, because they don't show any reference simulations, or measures of balance. It is surprising that their technique works so well, as there are publications that claim the opposite, i.e. not easy to create such balance in networks.

2) What's more, they present "tight balance" as if it exists in a vacuum, ("Also, it has been shown that this connectivity can be learned using simple spike timing-dependent plasticity rules (W. Brendel, R. Bourdoukan, P. Vertechi, et al., unpublished observations; Bourdoukan et al. (2012)), so extensive fine-tuning is not required to obtain these spiking networks.") but there are at least four other published methods of creating "tight" / detailed / precise" balance in recurrent networks: Haas et al., JNeurophys. 2006, Vogels et al., Science, 2011, Luz & Shamir, PlosCB, 2012 and Hennequin et al., Neuron, 2015. It's pretty powerful that the authors can just derive their architecture as they should be. To make this more interesting (and clear), the authors need to discuss and compare their result to these other studies. What are the crucial differences of "tight balance" to these other balances?

3) Another issue is the seemingly carefully calibrated "spike reset threshold", that makes sure that the inhibitory activity is equal to x^? How does that work, and why is this a plausible assumption, or is this just a trick to make sure the inhibitory input is equal to the desired output? Please explain.

4) Finally, they write "To be biologically more plausible, however, the derivative could be computed through a simple circuit that combines direct excitatory signal inputs with delayed inhibitory signal inputs (e.g. through feedforward inhibition)." It is not clear that FF inhibition would do this, because it crucially misses the translation of the dynamics to be ==x^. In previous work, it has been shown that deleting cells out of a FF network has immediate effects on the dynamics of the read out (Vogels, NN09, Figure 4). This may be a different scenario, but it is far from clear in the Methods, and if this constraint is crucial, it needs to be carefully discussed to avoid the impression that the paper at hand is inherently circular.

---

## [Author Response]

*All of the reviewers felt that the submitted theoretical work on compensating for cell death was novel and interesting. However, it was also felt that many specifics were unclear or buried in other places, making biological interpretation difficult (e.g., excitatory, inhibitory cell types). The reviewers encourage the authors to consider the five following points, and at least the first four should be addressed in a revised submission. Additional points raised by the reviewers are detailed below, and should also be addressed.*

*1) State clearly that the network is non-Dalian and explain the reasoning for this choice.*

The reviewers point out a fundamental omission in our original work: the difference between excitatory and inhibitory neurons, and their (potentially) contrasting effects on a network’s response to neuron loss. We have addressed this problem in the following way:

First, we have added a Dalian network to our work, following the recipe of Boerlin et al. (2013), and use this model now as a starting point. Second, we now clearly state when we use non-Dalian networks. Third, we explain the reason for using non-Dalian networks. In a nutshell, using non-Dalian networks does not change our predictions, but makes life simpler. More specifically, whether we take the Dalian (E/I) network proposed in Boerlin et al. 2013, or we take a non-Dalian network, the essential predictions about optimal compensation and recovery boundary remain the same. The key reason is that both networks in effect minimize the same objective function, and, since compensation effects are guided by the objective function, both networks show the same compensatory responses. This is now clearly explained in the Results and Methods sections – see also point 3 below. Given this equivalence, we use the simpler non-Dalian network because it removes one layer of complexity, which we hope aids the reader in understanding (or simulating) the networks.

*2) State clearly what the recovery boundaries are, and plot figures that show which variables affect the recovery boundary for a given network with a given task. In general, spend a bit more time on the general mechanism before going into the additional models, or alternatively, explain in detail the additional models, like e.g. Figure 7, which is devoid of explanatory power.*

We now explain the recovery boundary in greater detail. Specifically, we changed these items:

A) We now explain both the general compensatory mechanisms and the recovery boundary in greater detail by splitting them into two separate figures (Figure 2 and Figure 3). Figure 2, which now contains a Dalian network, explains compensation in the simplest possible case (one input signal in a network of 100 neurons). Figure 3 explains compensation and the recovery boundary in greater detail, and specifically shows (1) the interplay of neural tuning, neuron loss, and compensation, and (2) the breakdown in E/I balance in part of the population as a function of input signals and neural tuning. The latter part highlights that at the recovery boundary, balance will generally break down in only part of a population, and for only part of the input signals.

B) We clearly explain the recovery boundary for both the network with monotonic tuning curves and the networks with bell-shaped tuning curves. Both networks have been condensed to a single figure (now Figure 5).

We also included a mechanistic subtlety we did not address in the previous manuscript, namely the difference between two definitions of E/I balance: a standard definition in terms of synaptic inputs, and a non-standard definition in which a neuron’s self-reset is considered part of the inhibitory inputs. In large and massively redundant networks, the difference can be neglected. In smaller networks, or large networks with limited redundancy (defined as number of input signals over number of neurons), the difference is important, and the non-classical definition more clearly captures what we call re-balancing or a break-down in balance.

*3) An example of a Dalian network, independently of whether it works (same boundary?) or not (why does it not work?) should be added.*

As stated above, using a Dalian network does not change the results. In a nutshell, the Dalian network has the same recovery boundary. To show that, and to be clearer about our use of non-Dalian networks, we now changed the order of the text, and we start with a Dalian network (new Figure 2). We furthermore added a section in the methods that explicitly explains how to construct Dalian networks that minimize an objective function, and that explains the equivalence, in terms of compensation effects and recovery boundaries, of Dalian and non-Dalian networks.

In short, here’s how we solve the problem of Dale’s law. First, we consider that all neurons simulated in the non-Dalian networks are excitatory, except that they have direct inhibitory connections with each other. In the Dalian network, these direct inhibitory connections are then re-routed through a separate population of inhibitory neurons. This population simply seeks to represent the firing of the excitatory neurons as faithfully as possible, using the same principle of efficient coding used in the excitatory population. As long as the inhibitory population is fully functional, it acts essentially identical to the direct inhibitory connections, and the compensatory properties of the excitatory neurons in the Dalian and non-Dalian networks are therefore the same. (‘Essentially’, because there can be minor quantitative differences due to approximation errors in smaller networks; as the network size grows, they become exactly the same.) The inhibitory population can similarly compensate for the loss of its neurons (up to a recovery boundary), and its compensatory properties are similar.

*4) Show the compensatory mechanisms at play more carefully in figures, i.e. show how balance is changing and finally breaking at the recovery boundary.*

As stated above, we have expanded the mechanistic explanations in the first figures (Figure 2 and Figure 3). We note that EI balance – when defined to include a neuron’s self reset – is not changing as long as the system can compensate for the loss of neurons. Once it is no longer able to compensate for certain signal dimensions, balance in all neurons coding for those signal dimensions breaks down. Importantly, the only measurable signature of successful compensation are therefore changes in firing rates. That is also one of the main reasons why the figures (now Figure 5–Figure 8) that discuss various implementations of the framework for different neural systems focus mostly on firing rate changes.

*5) Consider building a reservoir network model as in Sussillo & Abbott trained to perform a task. Delete neurons, and check whether it is able to self-balance. If it doesn't, address whether this is related with the complexity of the task.*

This is an interesting suggestion, and we believe it is indeed important to evaluate how different types of network models respond to neural damage. We have opted not to include this in this paper for two reasons. First, we feel that our paper, as it stands, is already quite full with materials, and we did not want to overburden the reader any further. Second, we think this point really deserves a wider investigation, i.e., should include models beyond the Susillo & Abbott model, to study their ability to compensate for neuron loss. A graduate student is currently actively working on this larger problem, and we intend to publish these results elsewhere.

Additional changes:Besides these major changes, we point out a few other changes in the main manuscript:

A) Based on a comment of reviewer #2, we found that our over-emphasis of ‘neuron death’ may mislead readers, so we now prefer to opt for the slightly more neutral term ‘neuron loss’. Accordingly, we changed the title from ‘Optimal compensation for neuron death’ to ‘Optimal compensation for neuron loss’.

B) The manuscript changed substantially, so we opted not to provide a version in which changes are tracked because that version looked too messy.

*We are including the original reviews as they provide context for the revision points shown above.*

*Reviewer #1:*

*In this clearly written model theory paper, the authors propose that neural circuits compensate for neuronal death immediately, and show this using firing rate network models and several experiments. Although overall I found the work compelling, in general, I felt that the authors need to better describe 'biological correlates' for their models and to 'fill in the blanks' in various places to help the reader relate and better appreciate the work.*

*Specifically:*

*1) 'quadratic programming algorithm' and 'loss function' – could the authors provide suggestions of biological correlates of this. Presumably it would relate to excitatory-inhibitory balances in some way?*

Throughout the manuscript, we have tried to de-emphasize mathematical concepts (e.g. quadratic programming and loss function) in favor of biological concepts. We have also expanded and rewritten the beginning of the Results section to explain these concepts more carefully.

In a nutshell, the key idea is the following: the patterns of spikes emitted by our networks ‘self-organize’ such that downstream readouts can extract the input signals from a highly efficient code. This self- organization is coordinated by the networks’ own internal dynamics and corresponds, mathematically speaking, to the minimization of an objective or loss function (which measures the quality of the population code). This idea of minimizing an objective with a network is quite similar in spirit to e.g. the Hopfield model. However, in our network we can actually identify a simple biological correlate of the loss function since each neuron’s membrane voltage measures a part of the coding error. Hence, information about the loss function is hidden away in the (subthreshold) membrane voltages, excitatory inputs cause increases in coding errors, and inhibitory inputs cause decreases in coding errors. If excitation and inhibition are balanced in every single neuron, then all coding errors are kept in check.

*2) While spiking rate models are used, differences between excitatory and inhibitory cells are not apparent to me. Given this, then the suggestion in the Discussion that 'these predictions could be tested using neuronal ablations or optogenetic silencing' would seem to suggest that cell type (at least as far as excitatory or inhibitory) does not matter, but this is clearly not true (one of many examples would be epilepsy work by Soltesz group – Krook-Magnuson et al. 2013 Nature Communications). For the reader to better appreciate what is intended by the various discussion suggestions, the authors should 'fill in the blanks' between their theoretical model and experimental suggestions being made.*

*In the Methods it is stated that "To be biologically more plausible, however, the derivative could be computed through a simple circuit that combines direct excitatory signal inputs with delayed inhibitory signal inputs (e.g. through feedforward inhibition)."*

*So, could the authors provide description/interpretation of excitatory/inhibitory cells/circuits in terms of their results/predictions? (i.e., biological plausibility).*

Thanks for pointing out this fundamental omission. It was certainly not our intention to suggest that cell types are irrelevant. To address this point, we now start with a network that explicitly distinguishes excitatory and inhibitory neurons. We show that for each of these populations similar compensatory effects apply. We explain the resulting compensatory boundaries, and then move to networks that leave out the inhibitory subpopulations for simplicity. We discuss the recovery boundary for the excitatory population, but we do not explicitly discuss what happens in our networks when the inhibitory population can no longer compensate neuron loss (the result is an explosion of activity, whose exact nature we did not investigate).

The work by the Soltesz group is really interesting, but we are not sure that it directly applies to our case. Rather than moving from an intact to a pathological case, Krook-Magnuson et al. move from the pathological to the intact state (through inhibiting a subset of cells) Our framework makes clear predictions for what should happen when a system compensates for neuron loss; once a system moves beyond the recovery boundary, we essentially only know that the representation has to partially break down, and we can predict the immediate nature of the errors. However, it is much harder to predict (and likely problem-specific) how these immediate errors will affect the subsequent dynamics of the system.

*3) The authors use only positive firing rates (unlike other theoretical studies with positive and negative etc.), and this is a critical difference in what they achieve. While it is clear that negative firing rates may not make sense, the biological correlate of the authors' model is not clear – see 2.*

The biological correlate is always a balanced, excitatory-inhibitory network, and we now explain this in greater detail at the beginning of the Results. To understand what is happening in these quite complex networks, we make two simplification steps. We first eliminate the inhibitory population and replace it by direct inhibitory connections between the excitatory neurons. We then show how one can understand the firing rates of the resulting networks via a mathematical optimization procedure (quadratic programming) that provides additional insights. Importantly, these are only mathematical simplifications, but not biological simplifications. In other words, all the results we obtain for the simplified networks or the firing rate calculations still apply to the general excitatory-inhibitory network. We now explain this latter bit in the Methods section.

*Reviewer #2:*

*[…] There are however some aspects that the authors need to address in order to present the limitations of the model and the comparison with the data more clearly (see Comments below).*

*Comments:*

*Dale's law and the explanation of the EI balance breakdown. In Figure 2, the authors illustrate how when a fraction of the neurons below the recovery boundary is inactivated, the network dynamically compensates and keeps encoding x(t): the remaining neurons change their firing rate and the EI balance is maintained. Then they show that when all neurons with positive readout weight are removed, the balance can no longer be maintained because "there are no longer enough inhibitory cells to balance excitation (or enough excitatory cells to balance inhibition)". This is a very misleading statement given that in this network there are no excitatory or inhibitory neurons (i.e. does not respect Dale's principle). I was very confused when I read this example thinking that neurons in the Figure should have been organized according to whether they are E or I, and not according to the sign of their readout weight. I had to go to the Methods section to later find out that there are no E and I cells in this network. This is in my opinion a main drawback given that (*1*) Dale's law seems to be true in cortical circuits and (*2*) the authors are presenting their work as an extension of classical work describing EI networks in which E and I populations are distinct. In their previous manuscript (Boerling et al. 2013) they address this question and propose potential solutions. All of it should be explicitly mentioned and discussed.*

The reviewer raises a valid and important point which we did not address in the original manuscript. We have now corrected this omission. Specifically, we now start the Results with a general EI network, following the core ideas of Boerlin et al., 2013, and we separately consider the compensatory properties of the excitatory and inhibitory populations. We then move on towards the non-Dalian networks, a step that is done purely for mathematical convenience, since it removes one layer of complexity (separate inhibitory neurons) without changing the results. Importantly, the compensatory mechanisms are the same for these EI networks compared to the simpler non-Dalian networks. We explain the equivalence of the Dalian and non-Dalian networks in the Methods section.

Concerning the unfortunate confusion resulting from the sign of the readout weights in the old Figure 2: we have now opted to eliminate this additional complexity from the figure, so that readers will not confuse the sign of the readout weights with excitation vs. inhibition.

*Moreover, when describing Figure 2 it should be explained why cells with positive readout weights (w+) can naively be viewed as excitatory cells when they provide both excitation and inhibition? Without this most readers will be misled to interpret that w+ neurons are Excitatory and w- neurons are inhibitory.*

Our description of these cells was indeed misleading, and we apologize for the resulting confusion. We have now replaced Figure 2 with a simulation of an EI network, and we no longer consider +/- neurons in order to simplify the explanations.

*Biased cost. The authors use a biased quadratic cost function containing a second term which tries to maintain a particular read-out of the rates (coefs c_k_) equal to a constant that they call the background rate. The authors describe in much detailed the implication of this bias in the intercept of the tuning curves (Figure 3 and Figure 3—figure supplement 2B-C). The motivation for this choice is however very succinct. Is there any benefit on having a non-zero intercept regarding the estimation error?*

The introduction of the biased cost is indeed unfortunate. We now eliminated the term ‘biased cost’ and provide a different explanation for it. Briefly, we assume that the respective systems represent two signals, one that is variable, and one that is constant, which we now call ‘background signal.’

Of course, these changes are cosmetic and do not explain why we include such a background signal in the first place. Our explanation is two-fold. The first reason is pedagogical. Without the background signal, the tuning curves of one-dimensional systems are not particularly interesting, so we need a second signal to obtain more interesting tuning curves. In some sense, this is the simplest system that has interesting (piecewise linear) tuning curves. The second reason is the oculomotor integrator. We re-interpret this system in the following way. Rather than assuming that the system represents eye position per se, we now assume that it seeks to represent the two muscle activities that steer (horizontal) eye movements. These two muscles are antagonists, meaning that they contract and relax in opposition to each other. Assuming that we call the muscle activations *m_L_
*and *m_R_*, then the eye position is, to first approximation, given by the difference in muscle activation, *x* = *m_R_ − m_L_*. At the same time, the sum of the two muscle activations, *y* = *m_R_
*+ *m_L_
*will remain somewhat constant, again to first approximation. This is obviously a simplification, but we think a mild one. This second variable, *y*, is then mathematically identical to the constant, ‘biased cost’ from the previous manuscript.

*I ask this because it is a fairly common observation that cortical circuits seem to maintain the overall average population firing rate constant when dynamically encoding different variables (see e.g. Figure 2 Fujisawa et al., Nat Neuroscience 11 (7), 2008). This observation seems to be obtained to the network in which there is a non-zero r_B_ and non-zero uniform c_k_'s which seems interesting. However, the implications of this additional bias regarding the encoding ability of the circuit (estimation error of x(t)), its plausibility regarding the available data, etc. and not discussed.*

The link to cortex is indeed quite interesting. However, our rewording of the oculomotor system in terms of muscle activities (as opposed to a biased cost) weakens that link a bit. There may still be a common underlying principle, but we don’t have a clear-cut, solid answer for what that would be.

*The prediction on the changes in tuning for V1 neurons is only shown in three units (Figure 8). In Figure 8—figure supplement 1, the full population of neurons is examined but the changes are only reported in response to the "preferred stimulus orientations". Thus, evaluating the prediction that there should be an increase of firing rate on the non-silenced neurons in response to the "preferred orientation of the silenced cells" is not possible. Either a different population analysis is performed or the statement about the outcome of the prediction at the population level (Results, last paragraph) should be modified.*

We now plot the tuning curves of the full population in the new Figure 8, and we show all changes in firing rates in the new Figure 8. The plots illustrate that many (but not all) neurons change their firing rates in response to the preferred orientations of the silenced cells.

Our initial choice of showing a few examples was largely motivated by comparing our results with the original experimental papers (by Crook and colleagues). This series of papers is based on describing the firing rate changes in single neurons, and has few summary statistics. We therefore kept the single neuron examples that we compare to the data, and moved them into the new Figure 9.

*Results on the comparison with data should be brought forward. Since the paper is quite long as it is now, and the most novel and most interesting part is the comparison of the model predictions with data from three systems (Figure 5–Figure 8), I suggest moving the mechanistic part of how to implement the optimal solution using a spiking network (e.g. Figure 1–Figure 2) to an Appendix box or some sort of "side note figure". This mechanistic part is an important aspect but, once it has been shown that it works, which has somehow already been covered in Boerling et al. 2013, it is no longer required to make the comparison with data (Figure 5–Figure 8, which are all derived by numerically solving Eqs. 4-5). Doing so, the reader will be able to quickly reach the most novel aspects without the need to go over the network implementation details.*

We would ideally like the paper to be self-contained and oriented towards biologists. While we agree that readers that have read and digested the paper by Boerlin et al. will find the mechanistic explanations potentially repetitive, we do not want to re-direct the more biological readership to the Boerlin paper. We therefore chose to keep the mechanistic explanations, but we sought to focus better on the compensation vs recovery boundary predictions of the model, and we also now explain an EI network. However, we did shorten the mid-section of the paper which concerned the computations of firing rates, and we provided additional simulations for the V1 model.

In terms of comparison with the data, we provide a more detailed account of the data in the main text. We emphasize that the original series of studies by Crook and Eysel is largely based on analyzing single neurons one by one, so we did not change our plots. Our V1 model captures the core qualitative aspects of the data (namely, that firing rates across the population change in the direction of the silenced neurons), but not every single aspect of it, which we now also explain in the Results section.

*In the Discussion, the authors state that the strongest prediction of the model is that "neurons with similar tuning to the dead neurons increase their firing rates and neurons with dissimilar tuning decrease their firing rates (unless they are already silent)." The second part of this statement seems at odds to what is shown in Figure 6, Figure 8 (top) and Figure 8 where dissimilar neurons either don't change much (e.g. Df is always >=0 in Figure 8) or tend to increase their rate in response to the preferred orientation of the dead cells (Figure 8 top). This rate decrease in dissimilar neurons seems to occur in the network shown in Figure 4 with monotonically increasing tuning curves. The details about the changes predicted and the generality of these changes in different systems (bell-shaped vs. monotonic tuning curves) need to be clarified.*

This is an excellent point, and has allowed us to uncover some mistakes and conceptual imprecisions in our previous version.

First a mistake: in the previous version, we used the word ‘orientation tuning’ (which goes from 0*−*180^*◦*^), where we should have used the word ‘direction tuning’ (which goes from 0 *−* 360^*◦*^). Indeed, the experiments by Crook and Eysel knock out sets of neurons with specific direction tuning (not orientation tuning). While we had done exactly that, it was not clear from the main text. This is now corrected.

Second, the conceptual problems: We had indeed written similar and dissimilar tuning, in order to emphasize a straight line of thought from the simple 1d and 2d models. However, this was simply not the right way to think about compensation in a high-dimensional system. The key problem is that ‘similar’ or ‘dissimilar’ tuning will usually be assessed with respect to a few parameters (such as the direction of a moving grating, e.g.). However, to understand compensation one has to assess tuning in the original, high-dimensional space, and compare the relative directions of the decoding weights of the knocked-out neurons with the direction of the stimulus, and the direction of the remaining neurons.

The core reason why neural firing rates increase to both the preferred and anti-preferred direction in the V1 model can therefore only be understood in this high-dimensional space. Briefly, most neurons are only weakly direction-tuned. Consequently, when we knock out a set of neurons with a given preferred direction, the network model experiences a strong representational deficit for stimuli of this preferred direction, and a weaker representational deficit for stimuli of the anti-preferred direction. Since none of the neurons are likely to directly point in the correct directions in the high-dimensional space, the representational deficit is most easily fixed by increasing the firing rates of a combination of the other neurons. While decreases in firing rates are also possible, they generally tend to have less of an impact.

We have reworded the complete V1 section in order to clarify these points. We have also added a schematic (new Figure 8) to illustrate graphically how compensation works in high-dimensional spaces.

*At the end of the Methods, the authors provide an explanation of the Tight balance between excitation and inhibition and a scaling argument that concludes that in the network consider here, the cancellation between the mean excitatory and inhibitory drives occurs with precision 1/N. They compare this cancellation with the classical balanced network presented in van Vreeswijk and Sompolinsky (1998) where the cancellation occurs up to order 1/sqrt(N). This comparison is interesting but in order to make it the authors should provide some more information about the behavior of the network in the large N limit: if x is constant and independent of N, do the firing rates converge to a constant value as N increases? This seems hard to visualize as, even when counting on the cancellation of the mean E and I inputs, the magnitude of the total input current fluctuations σ will grow with respect to the threshold (σ~1/N^2/3 whereas threshold is 1/N^2). This is a reflection of the fact that the rate of incoming spikes grows with N whereas the size of each PSP is always of the order of the threshold (i.e. 1/N^2). I don't see how this network can asymptotically converge to non-zero rates in this large N limit.*

The section on scaling at the end of the Methods was indeed quite brief, and we have now added more details. Concerning the reviewer’s question, we note that a crucial point in scaling up net- works is which variables change and which remain constant. When we scale up our networks, we keep two quantities fixed. First, we assume that the readout (*x*ˆ) stays the same. Second, we assume that the average firing rates of the neurons (*ri*) stay the same. In other words, the firing rates converge to constant values more or less by definition, or by our choice of scaling. Given that the signal and the firing rates remain constant, everything else has to scale. So it follows that the decoders *Di* have to scale with 1*/N*

(since *x*ˆ = ∑*_i_ D_i_r_i_*), and from that follows the scaling of the connectivities and currents etc. This is now more clearly explained in the Methods.

*A final comment: the authors present as an advantage the speed of the compensation mechanism which they say is "faster than the timescale of neural spiking". To me this comment remarks the mismatch of time-scales between the two processes: (1) neural death, an infrequent event, relative to the time scale of neurons, and (2) balanced dynamics, almost instantaneous. More than a bonus I see it as an indication that there are probably other slower processes at work when compensating for such an irreversible damage that happens in such long time-scales. Presented as a prediction for manipulation experiments, the compensation can be viewed as the signature mechanism of balanced dynamics, as done here in the comparison with the experimental data. Additionally, there might be circumstances in which groups of neurons become transiently and frequently inactive (e.g. local DOWN state). The fast compensation described here would seem more tuned to address those fast inactivation events instead of neural death.*

That’s an excellent observation and we totally agree. Indeed, there are likely many compensation mechanisms, and the slower ones that have to do with adapation, learning, plasticity etc. are certainly much better investigated. We briefly mention this at the beginning of the Discussion. However, the reviewer’s observation has also helped us to note more clearly the mismatch between the title (‘neuron death’), the inactivation experiments we describe, and the instantaneous compensation we advocate. With hindsight, we believe that the emphasis on ‘neuron death’ may in fact be misleading for potential readers. Hence, we decided to change the title and wording to a slightly more neutral term: ‘neuron loss’.

*Reviewer #3: […] Here, Barret et al. pick up from the well-described recurrently balanced neural network literature and study how balanced networks of various sizes compensate for neuron loss. They set out with two simple assumptions, i.e. that (1) outside stimuli can be described by combinations of quasi-steady-state firing rates of all neurons in the network and that (2) these firing rates are bounded by a cost function that prevents combinatorial codes in which the majority of neurons are silent in favour of a few active ones. With these two simple ingredients the authors show that the total number of spikes remains constant in the face of dramatic cell losses, as long as a tight balance of excitation and inhibition in the remaining neurons can be maintained. In fact, it is this balance that, when disturbed by cell loss, increases or decreases the firing rates in the remaining neurons to maintain faithful cumulative signal representation. The authors go on to show that this simple feature of balanced networks can account for compensatory behaviour in more complicated models such as Ben Yishai and Sompolinsky's (1995) bump attractor model (that is unfortunately not cited) and even more laborious models of sparse coding in V1. The results evoke vacillating responses in this reviewer of alleging outrageous triviality (anyone modelling balanced systems intuitively knows that this compensation happens) and respecting the originality of the thought process behind the sequence of presented results. After some deliberation, I agree with the authors fully in that optimal compensation by means of shifting, detailed balance has not been proposed before and is a worthwhile idea to entertain.*

We thank the reviewer for the helpful and honest comments. We totally agree that the rebalancing of balanced networks is, in many respects, a trivial property, and we now emphasize this a bit more in the Discussion, and seek to contrast this purely dynamical property of balanced networks with the notion of compensation, which implies a specific function. We also added a citation to the bump attractor model – thanks for pointing out this omission.

*To show the link of shifting balanced dynamics in response to cell death, I would like to ask the authors to include better explanations and figures panels for all models that exhaustively show the mechanistic origins of the phenomenon they showcase, and not just its effects, i.e. the final tuning curves before and after. Additionally, I would wish for a better graphical way to discern between the two cases of cell loss on either side of the recovery boundary.*

We agree with the reviewer’s suggestion. We have now expanded the Results section in order to explain the compensatory mechanism and its breakdown better. Specifically, we now included an EI model (new Figure 2), and we included a bell-shaped tuning curve model (Figure 3) in which we explain in detail the compensatory mechanisms before and after the recovery boundary has been hit. We emphasize that the main experimental signature of successful compensation is a change in firing rate with EI balance remaining intact and essentially unchanged, whereas the main experimental signature of a failure to compensate is a breakdown in balance. We now explain that this breakdown in balance is signal-dependent and only occurs in part of the population.

*In my opinion, Figure 3, Figure 4, Figure 5 could be integrated into Figure 2 or outsourced to supplementary materials and give way for the interesting cases of Figure 6 without much loss of detail.*

We have streamlined the presentation around Figure 3–Figure 5. Specifically, we have condensed and integrated the old Figure 3 and Figure 6 into the new Figure 5. In turn, we have expanded the beginning in order to emphasize the mechanistic origin of the compensatory effects, and we have expanded the section around the experimental results, and specifically the V1 model, which is now explained in much more depth.

*Lastly, I would like to ask the authors to discuss clearly what experimental results could discriminate the initially mentioned scenarios A and B (redundancy and compensation, respectively), and what they would expect in the case of temporally varying neuronal representations such as the en vogue neuronal dynamics of Churchland and Shennoy (2012).*

We emphasize that some redundancy is always necessary to compensate for neuron loss – without any redundancy, there can be no compensation. However, redundancy by itself is not sufficient.

Scenario A is redundancy without compensation. In this case, there would be degradation of performance, essentially proportional to the number of neurons lost. In real systems, one may expect that plasticity mechanisms in turn correct for that degradation, albeit on a slower time scale than the instantaneous correction proposed in our work.

Scenario B is (at least some) redundancy with compensation, meaning that firing rates in the remaining neurons actively adjust in order to compensate for the lost neurons. In this case, there should be no degradation whatsoever – until the recovery boundary is hit. Figure 9 now provides additional graphical intuition about what optimal compensation does.

In short, the best test would be an instantaneous (e.g. optogenetic) knock-out of a significant fraction of a network’s population. If the firing rates of the neurons adjust immediately so as to restore a (previously determined) readout, then some form of optimal compensation is at work. If the firing rates do not adjust, then the readout will be compromised to some extent, in which case we are just witnessing redundancy. Of course, the firing rates could also change in a way that is detrimental to a given readout, which could be a scenario C, a system that is not at all prepared to deal with neuron loss. Classical (non-robust) models of line attractors would probably fall in this latter category.

Concerning temporally varying representations, such as those of Churchland and Shenoy, our predictions would remain the same. A smaller fraction of knocked-out neurons could be compensated for, and a time-varying representation would remain intact. (In fact, this is exactly what happens when we take the slow connectivities from the Boerlin 2013 paper back into account.) Once the recovery boundary is hit, the representation would fail. In a dynamical system with time-varying representations, the resulting errors could then potentially build up, but studying the precise build-up of those initial representation errors would depend on the specifics of the underlying network architecture. (We are currently investigating some of these scenarios in detail, in order to better account for forthcoming experimental data; we chose not to include these results here, given that the current study is already quite lengthy.)

We discuss several possible experimental tests of our theory in the Results section. However, the key test is the one explained above: measure the instantaneous firing rate changes after partial knock-down of a neural population, and verify that these changes restore a (previously established) linear readout.

[Editors' note: further revisions were requested prior to acceptance, as described below.]

*The authors have addressed four of the five suggestions we made in the first round of reviews, and the paper has improved. However, the fundamental logic of the paper is still not quite clear and needs to be made transparent. Specifically, the Methods section is hard to follow, and there seem to be some discrepancies in updating (e.g., it refers to a Figure 2 that does not exist, and Figure 2 is not referred to in the text). The derivation and translation into Dalian spiking networks is far from clear.*

We thank the reviewers for their comments. Here is what we did to address them:

Methods: We now explain the derivation of the Dalian spiking network in much more detail, and we have eliminated some inconsistencies in the notation that may have caused confusion. Furthermore, we rewrote the section about our definition of EI balance.

Results: We have similarly rewritten the parts concerned with the Dalian networks and the explanation of EI balance, i.e., the text around Figure 2 and Figure 3.

Figure References: We have systematically checked all figure references, and eliminated a few typos and inconsistencies, including those mentioned.

Code: We have included the MATLAB code for all figures in this revision, which will allow both the referees and future readers to replicate and modify the simulations that we show.

*Further, please note the following four points:*

*1) They create what they call "tightly balanced" networks, i.e. networks with very fast, precise / detailed inhibition by means of mathematical derivation and go on to say that the networks they built are much more balanced than vanVreesvijk models. We cannot assess this, because they don't show any reference simulations, or measures of balance. It is surprising that their technique works so well, as there are publications that claim the opposite, i.e. not easy to create such balance in networks.*

We agree that we did not give a detailed comparison of balance in our networks compared with those of van Vreeswijk & Sompolinsky. Briefly, when increasing network size, balance scales with

1*/N* in our networks, rather than 1N as in the networks of van Vreeswijk & Sompolinsky. Hence, for larger networks, our networks are much more tightly balanced.

We did not provide a detailed comparison for two reasons. First, we had previously published the scaling arguments (Boerlin, Machens, Deneve, 2013, PLOS CB), and we believe that the brief recap in the Methods section is therefore sufficient. Second, we recently wrote a review (Deneve & Machens, 2016, Nature Neuroscience) that explains many of those differences on a more conceptual level. However, we agree that those articles were not properly referenced, and we now explicitly included references to tight/loose balance in the method section (‘Tight balance of excitation and inhibition’).

While we did not explicitly study or simulate other balanced networks, we do show measures of EI balance in our networks in Figure 2 and Figure 3. To better explain these measures, we have expanded and rewritten the section ‘Tight balance of excitation and inhibition’ in the Methods.

We are not entirely sure who argued that it should be difficult to create such a tight balance in networks. To show that doing so is rather straightforward and does indeed work well, and to allow readers to easily replicate our simulations and manipulate them, we included the complete simulation code (in MATLAB) for all figures.

*2) What's more, they present "tight balance" as if it exists in a vacuum, ("Also, it has been shown that this connectivity can be learned using simple spike timing-dependent plasticity rules (W. Brendel, R. Bourdoukan, P. Vertechi, et al., unpublished observations; Bourdoukan et al. (2012)), so extensive fine-tuning is not required to obtain these spiking networks.") but there are at least four other published methods of creating "tight" / detailed / precise" balance in recurrent networks: Haas et al., JNeurophys. 2006, Vogels et al., Science, 2011, Luz & Shamir, PlosCB, 2012 and Hennequin et al., Neuron, 2015. It's pretty powerful that the authors can just derive their architecture as they should be. To make this more interesting (and clear), the authors need to discuss and compare their result to these other studies. What are the crucial differences of "tight balance" to these other balances?*

It was certainly not our intention to slight the literature here, and pretend that we are the first to generate tightly balanced networks. We added a paragraph in the Discussion about ‘tightly balanced networks’ and included the mentioned references.

To address the specific point raised, we note that not all tightly balanced networks are the same. Apart from different assumptions about the underlying neuron models, and different definitions of balance, the network models also have quite different connectivities. We included a reference to an (as of yet) unpublished paper only because we wanted to address a reader’s potential worry that the connectivity in our networks is essentially designed (and not random as in most balanced networks).

More generally speaking, and referring back to point (1) as well, we absolutely agree that the commonalities and differences between our balanced networks and other balanced networks need to be worked out in greater detail. In fact, we are currently working on this (together in collaboration with Alfonso Renart – who has developed yet another tightly balanced networks). However, working out precisely which differences are important and which are unimportant is essentially a full scientific study in itself. While we appreciate and share the reviewer’s concern about understanding better how other types of networks relate to our framework, or how they may respond to neuron loss, we think that these topics are beyond the boundaries of the current paper, whose main focus is compensation for neuron loss, and its effect on neurons’ tuning.

*3) Another issue is the seemingly carefully calibrated "spike reset threshold", that makes sure that the inhibitory activity is equal to x^? How does that work, and why is this a plausible assumption, or is this just a trick to make sure the inhibitory input is equal to the desired output? Please explain.*

At first sight, it may seem that each neuron has a precisely tuned threshold and reset, which are coupled to the connectivity via the decoder weights. However, there is no fine tuning or careful calibration necessary. To see that, we first note that we can set all thresholds to the same value by redefining the voltages of the neurons (see e.g. Boerlin et al. 2013, PLOS CB, where this exercise is carried out in the subsection ‘Scaling and Physical Units’, Eq. (12)-(14)). After rescaling, the voltage reset is simply given by the equation

reset = −threshold − cost.

Accordingly, the reset does not need to match the (negative) threshold. Rather, any mismatch between the two can be interpreted as a cost term in the loss function. Such cost terms – as long as they are not excessive – will change the population spike patterns, but not the function of the system. Accordingly, we do not need to calibrate threshold and reset, rather, any mismatch will simply change the cost. Changing the cost, in turn, does change the precise spiking pattern of the system, but not its functionality and not its robustness to neuron loss.

*4) Finally, they write "To be biologically more plausible, however, the derivative could be computed through a simple circuit that combines direct excitatory signal inputs with delayed inhibitory signal inputs (e.g. through feedforward inhibition)." It is not clear that FF inhibition would do this, because it crucially misses the translation of the dynamics to be ==x^. In previous work, it has been shown that deleting cells out of a FF network has immediate effects on the dynamics of the read out (Vogels, NN09, Figure 4). This may be a different scenario, but it is far from clear in the Methods, and if this constraint is crucial, it needs to be carefully discussed to avoid the impression that the paper at hand is inherently circular.*

We believe that this is a misunderstanding. This section is simply about the biophysical interpretation of the external inputs into the recurrent network. The section does not contain any additional constraints or assumptions about the network model.

To clarify, we note that each neuron receives external inputs of the general form ‘*x* + *dx/dt*’, i.e., a combination of the original signal, *x _j_*, and its derivative, *dx_j_/dt*. There are two ways in which we can interpret this external input biophysically. First, we can simply think of this as a single external current input *c(t*) with *c* = *x* + *dx/dt*. In this case, the signal *x(t*), which we use throughout the paper, is simply a filtered version of an actual external input signal *c(t*). Second, we can think of this as a biophysical computation that operates on the signal *x(t*). In this case, each neuron receives two types of external inputs, the original signal input, *x _j_ (t*), and its derivative, *dx_j_/dt*. The latter could be computed through a combination of direct excitatory signal inputs with delayed inhibitory signal inputs – and this is what we were referring to.

We have rewritten this section to explain the logic better and avoid any confusion.